# TIMELESS-TIPIN and UBXN-3 promote replisome disassembly during DNA replication termination in *Caenorhabditis elegans*

Yisui Xia ⬥, Ryo Fujisawa ⬥, Tom D Deegan ⬥, Remi Sonneville* ⬥ & Karim P M Labib** ⬥

## Abstract

The eukaryotic replisome is rapidly disassembled during DNA replication termination. In metazoa, the cullin-RING ubiquitin ligase CUL-2$^{LRR-1}$ drives ubiquitylation of the CMG helicase, leading to replisome disassembly by the p97/CDC-48 "unfoldase". Here, we combine *in vitro* reconstitution with *in vivo* studies in *Caenorhabditis elegans* embryos, to show that the replisome-associated TIMELESS-TIPIN complex is required for CUL-2$^{LRR-1}$ recruitment and efficient CMG helicase ubiquitylation. Aided by TIMELESS-TIPIN, CUL-2$^{LRR-1}$ directs a suite of ubiquitylation enzymes to ubiquitylate the MCM-7 subunit of CMG. Subsequently, the UBXN-3 adaptor protein directly stimulates the disassembly of ubiquitylated CMG by CDC-48_UFD-1_NPL-4. We show that UBXN-3 is important *in vivo* for replisome disassembly in the absence of TIMELESS-TIPIN. Correspondingly, co-depletion of UBXN-3 and TIMELESS causes profound synthetic lethality. Since the human orthologue of UBXN-3, FAF1, is a candidate tumour suppressor, these findings suggest that manipulation of CMG disassembly might be applicable to future strategies for treating human cancer.

**Keywords** CDC-48; CMG helicase; CUL-2$^{LRR-1}$; DNA replication termination; TIMELESS-TIPIN; UBXN-3

**Subject Categories** DNA Replication, Recombination & Repair; Post-translational Modifications & Proteolysis

The EMBO Journal (2021) 40: e108053

## Introduction

Eukaryotic chromosomes are copied just once per cell cycle (Bell & Labib, 2016; Burgers & Kunkel, 2017; Gasser, 2019), dependent upon a dynamic molecular machine known as the replisome (Bai *et al*, 2017). During S-phase, the replisome assembles around the CMG helicase at nascent DNA replication forks (CMG is named after its three sub-assemblies, namely the CDC-45 protein, the hexameric MCM-2-7 motor that encircles DNA and the GINS complex). After initiation, CMG associates continuously with DNA replication forks throughout elongation (Labib *et al*, 2000), until termination occurs when two DNA replication forks from neighbouring origins converge, or when a single replisome arrives at a telomere or DNA nick (Maric *et al*, 2014; Moreno *et al*, 2014; Dewar *et al*, 2015; Vrtis *et al*, 2021).

Work with budding yeast and metazoa indicates that the CMG helicase is ubiquitylated on its MCM7 subunit during termination (Maric *et al*, 2014; Moreno *et al*, 2014; Dewar *et al*, 2017; Sonneville *et al*, 2017). This leads to recruitment of the Cdc48/p97/ VCP ATPase via its UFD1-NPL4 adaptor proteins (Franz *et al*, 2011; Maric *et al*, 2017; Mukherjee & Labib, 2019; Deegan *et al*, 2020), which recognise polyubiquitin chains that are linked via lysine 48 of ubiquitin (Bodnar & Rapoport, 2017, Twomey *et al*, 2019, van den Boom *et al*, 2016). Cdc48/p97 then unfolds ubiquitylated MCM7 (Deegan *et al*, 2020), leading to the irreversible dissociation of CMG into its component parts and thus to replisome disassembly and the dissociation of replisome components from DNA.

Although metazoa and yeast share common principles of replisome disassembly during DNA replication termination, important differences are also apparent. Disassembly of the budding yeast replisome has been reconstituted with purified proteins, showing that the cullin 1 ligase SCF$^{Dia2}$ directs a single E2 ubiquitin-conjugating enzyme called Cdc34 to initiate and then elongate a long K48-linked ubiquitin chain on CMG-Mcm7 (Maric *et al*, 2014; Deegan *et al*, 2020). However, SCF$^{Dia2}$ is absent in metazoa. Work with the nematode *Caenorhabditis elegans* and the frog *Xenopus laevis* has shown that a cullin 2 ligase called CUL-2$^{LRR-1}$ is recruited to the terminating replisome and is required for CMG disassembly during termination (Dewar *et al*, 2017; Sonneville *et al*, 2017). The mechanism of CMG ubiquitylation by CUL-2$^{LRR-1}$ has yet to be determined in any metazoan species and until now the reaction had not been reconstituted *in vitro*.

The very high efficiency of CMG disassembly in budding yeast is enforced by two core replisome components called Ctf4 and Mrc1, which jointly recruit SCF$^{Dia2}$ to the replisome and thereby ensure that every CMG helicase complex is ubiquitylated during

The MRC Protein Phosphorylation and Ubiquitylation Unit, School of Life Sciences, University of Dundee, Dundee, UK
*Corresponding author. Tel: +44 1382 386395; E-mail: r.sonneville@dundee.ac.uk
**Corresponding author. Tel: +44 1382 384108; E-mail: kpmlabib@dundee.ac.uk

termination (Maculins *et al*, 2015; Deegan *et al*, 2020). In contrast, a potential role for metazoan core replisome components in stimulating CMG helicase ubiquitylation by CUL-2^(LRR-1) during DNA replication termination had not previously been explored.

Upon ubiquitylation of yeast CMG, the ubiquitin receptors Ufd1-Npl4 recruit Cdc48 and support helicase disassembly, provided that at least five ubiquitin moieties have been conjugated to CMG-Mcm7 (Mukherjee & Labib, 2019; Deegan *et al*, 2020). Although both yeast Cdc48-Ufd1-Npl4 and human p97-UFD1-NPL4 are sufficient to unfold a model substrate comprising a poly-ubiquitylated fluorescent protein (Blythe *et al*, 2017; Bodnar & Rapoport, 2017; Pan *et al*, 2021), work with *C. elegans* indicated that the disassembly of ubiquitylated CMG helicase by metazoan p97-UFD1-NPL4 is more complicated than would have been predicted by the corresponding studies of their yeast orthologues. A further adaptor of CDC-48/p97 called UBXN-3 was shown to contribute to chromatin unloading of CMG components in worms with reduced expression of CDC-48 (Franz *et al*, 2016). Moreover, UBXN-3 was found to be required for a second pathway of CMG disassembly that is activated during mitosis (Sonneville *et al*, 2017). This mitotic CMG disassembly pathway requires the TRUL-1/TRAIP ubiquitin ligase and helps to process sites of incomplete DNA replication (Deng *et al*, 2019; Priego Moreno *et al*, 2019; Sonneville *et al*, 2019). Until now, it was not known whether the role of UBXN-3 in the disassembly of ubiquitylated CMG by CDC-48_UFD-1_NPL-4 was direct and the reaction had yet to be reconstituted with purified proteins. In addition, it was unclear whether UBXN-3 also acts during DNA replication termination to stimulate CMG helicase disassembly by CDC-48_UFD-1_NPL-4.

Here, we use *C. elegans* as a model system to explore the mechanism of replisome disassembly during DNA replication termination by metazoan CUL-2^(LRR-1) and CDC-48_UFD-1_NPL-4. Our data show that the core replisome factors TIMELESS-TIPIN help to recruit CUL-2^(LRR-1) to the CMG helicase, in order to promote efficient ubiquitylation of CMG-MCM-7. Subsequently, UBXN-3 directly stimulates the disassembly of ubiquitylated CMG by CDC-48_UFD-1_NPL-4, not only during mitosis but also during DNA replication termination. Lack of both TIMELESS-TIPIN and UBXN-3 causes a synthetic defect in CMG disassembly both *in vitro* and also in the *C. elegans* early embryo, with the latter defect being associated with a profound loss of viability.

# Results

## An RNAi screen for E2 ubiquitin-conjugating enzymes that work with CUL-2^(LRR-1) in *Caenorhabditis elegans*

Relatively little is known about the mechanism of *C. elegans* cullin ligases and the identity of their cognate E2 enzymes. Nevertheless, studies of the equivalent human enzymes have shown that metazoan cullin ligases are considerably more complex than their yeast counterparts and function together with a complex array of different enzymes, in order to synthesise K48-linked ubiquitin chains on their substrates (Baek *et al*, 2020b; Wang *et al*, 2020). Firstly, the cullin scaffold must be modified by the ubiquitin-like protein NEDD8, which serves as a nexus that contacts multiple elements of the ligase along with the cognate E2 ubiquitin-conjugating enzyme (Baek *et al*, 2020a; Wang *et al*, 2020). Subsequently, specialised "priming" enzymes are responsible for the initial mono-ubiquitylation of substrate lysines, whereas distinct E2 enzymes then mediate the subsequent elongation of K48-linked ubiquitin chains (Kleiger & Deshaies, 2016). Recent work identified two different classes of priming enzymes for human cullin ligases (Fig 1A). The first comprises paralogues of the E2 enzyme UBE2D, which is activated by the RING subunit of a neddylated cullin ligase (Baek *et al*, 2020a). The second type of priming enzyme is an RBR ("RING-between-RING") E3 ligase of the ARIADNE family, known as ARIH1, which associates with neddylated cullin ligases and receives ubiquitin from the cysteine-specific E2 enzyme UBE2L3, before transferring this ubiquitin to a substrate lysine (Scott *et al*, 2016; Horn-Ghetko *et al*, 2021). Subsequently, K48-linked chains are extended on the primed substrate by the human orthologues of yeast Cdc34, known as UBE2R1-2, but these act redundantly with a further E2 enzyme (Fig 1A) called UBE2G1 (Hill *et al*, 2019).

The single *C. elegans* orthologue of UBE2D (LET-70) is essential for worm viability (Zhen *et al*, 1996), as is the LRR-1 substrate adaptor of CUL-2^(LRR-1) (Merlet *et al*, 2010). In contrast, we found that deletion of the sole orthologue of mammalian UBE2R1/R2 in *C. elegans* was viable (*ubc-3*, Appendix Fig S1A–B and G–H), as was deletion of worm UBE2G1 (*ubc-7*, Appendix Fig S1C–D and G–H), or mutation of worm UBE2L3 (*ubc-18*) at a site predicted to abrogate its interaction with E1 (Fay *et al*, 2003). To investigate which of the *C. elegans* E2 enzymes might function *in vivo* with

**Figure 1. An RNAi screen for candidate E2 enzymes that contribute to CMG-MCM-7 ubiquitylation during DNA replication termination in *C. elegans*.**

A Model for the priming and elongation of ubiquitin chains on substrates of cullin ubiquitin ligases in metazoa. See text for details.

B Family tree for the E2 ubiquitin-conjugating enzymes encoded by the *C. elegans* genome. For each of the indicated groups, a single plasmid was generated to express RNAi to the component genes (see Materials and Methods).

C Summary of RNAi screen to detect synthetic lethality, upon combining *ubxn-3* RNAi with a plasmid expressing RNAi to *lrr-1* or to one of the groups of E2 enzymes indicated in (B).

D Summary of synthetic lethality data for the screen described in (C). Worms were fed on the indicated proportions of bacteria expressing RNAi to *ubxn-3*, *lrr-1*, E2 enzyme groups G1-G6, or else containing empty vector as indicated. The data represent the means and standard deviations from three biological replicates.

E, F Synthetic lethality resulting from the combination of RNAi to E2 group 5 and RNAi to *ubxn-3* was deconvolved in similar experiments to those in (D), using plasmids expressing RNAi to individual E2 enzymes, or pairs of E2 enzymes, as indicated.

G *GFP-psf-1* worms were fed on bacteria containing a single plasmid expressing the indicated RNAi treatments, before preparation of embryonic cell extracts and isolation of GFP-PSF-1 by immunoprecipitation. The indicated factors were monitored by immunoblotting.

H The presence of GFP-PSF-1 on mitotic chromatin (indicated by white arrows) was monitored by spinning disc confocal microscopy (see Materials and Methods), in *GFP-psf-1 mCherry-Histone H2B* worms that were fed on bacteria containing a single plasmid expressing the indicated RNAi treatments ("Control" = empty vector). NEB = nuclear envelope breakdown. The scale bars correspond to 5 μm.

I Analogous experiments to those in (D, E) to deconvolve the synthetic lethality induced by RNAi to *ubxn-3* in combination with E2 group 6.

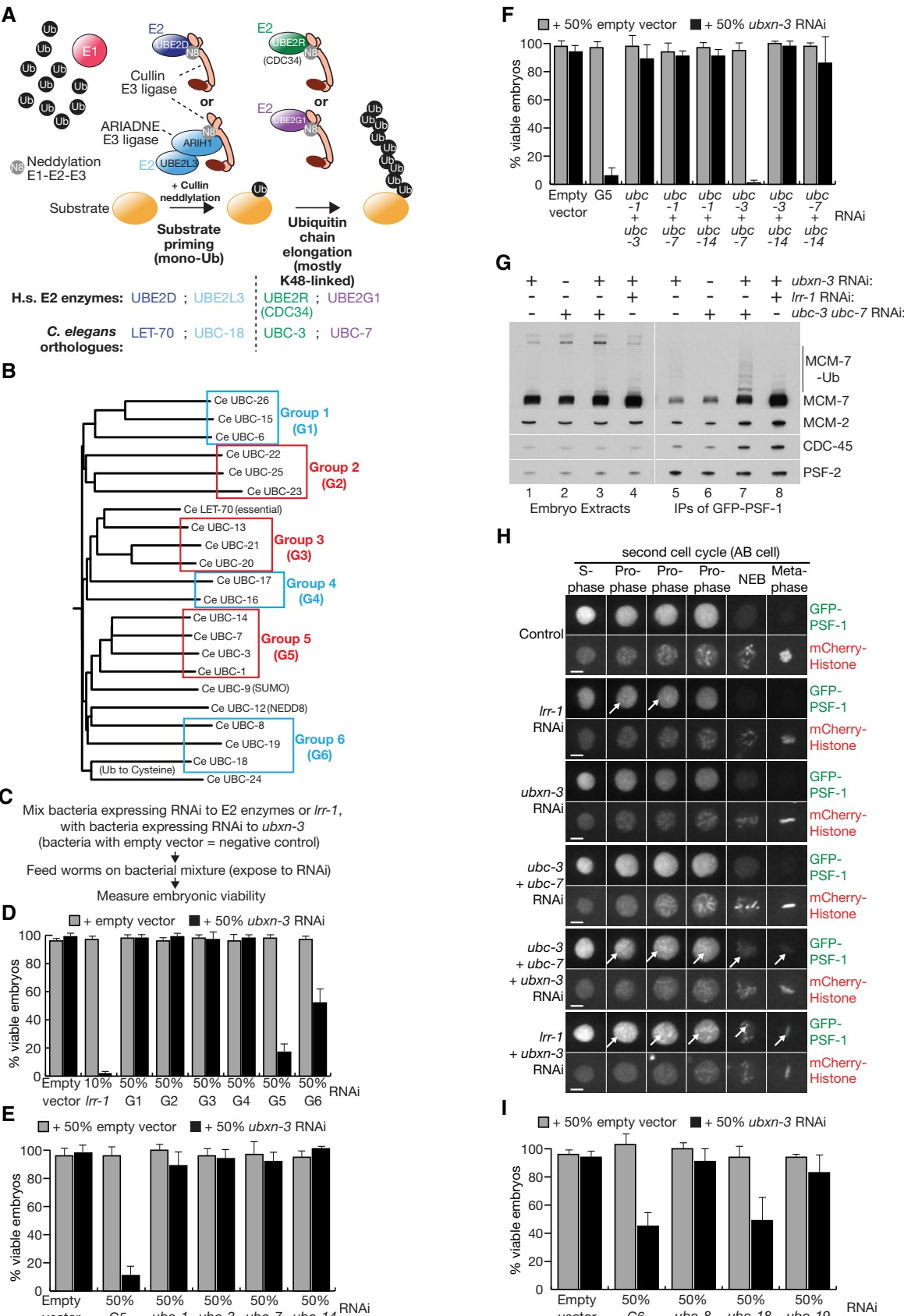

**Figure 1.**

CUL-2$^{LRR-1}$ during DNA replication termination, we developed a genetic screen that was based on our previous observation that partial RNAi inactivation of *lrr-1* is synthetic lethal with RNAi depletion of UBXN-3 (Sonneville *et al*, 2017). This indicated that RNAi depletion of E2 enzymes that act with CUL-2$^{LRR-1}$ during DNA replication termination might also cause synthetic lethality in combination with *ubxn-3* RNAi.

We divided the worm E2 enzymes into a series of phylogenetic groups and constructed RNAi plasmids to inactivate each group simultaneously (Fig 1B). We then fed worms on bacteria expressing *ubxn-3* RNAi or empty vector (Fig 1C), mixed one-to-one either with bacteria expressing each of the six groups of E2 RNAi, or with positive and negative controls (10% *lrr-1* RNAi and empty vector, respectively).

Most of the tested E2 groups had no impact on viability together with *ubxn-3* RNAi, whereas the combination of *lrr-1* and *ubxn-3* RNAi reduced viability close to zero (Fig 1D). However, E2 group 5 produced a strong synthetic lethal phenotype in combination with *ubxn-3* (Fig 1D), which subsequent deconvolution showed was dependent upon the combined inactivation of *ubc-3* and *ubc-7* (Fig 1E and F). Triple RNAi inactivation of *ubxn-3*, *ubc-3* and *ubc-7* led to the partial accumulation of the CMG helicase with short ubiquitin chains on MCM-7 (Fig 1G), and the partial retention of CMG on chromatin during mitosis (Fig 1H). Moreover, co-depletion of UBC-3 and UBC-7 reduced the ubiquitylation of CMG-MCM-7 in worms that had been treated with *npl-4* RNAi in order to block CMG disassembly by CDC-48 (Appendix Fig S2). These findings indicated that UBC-3 and UBC-7 contribute to CMG ubiquitylation and disassembly in the *C. elegans* early embryo, acting redundantly with each other.

Whereas deletion of *cul-2* or *lrr-1* is lethal in *C. elegans* (Feng *et al*, 1999; Merlet *et al*, 2010), the combination of *ubc-3*Δ and *ubc-7*Δ is viable although the brood size is reduced (Appendix Fig S1G and H). This indicates that other E2 enzymes must be able to act with CUL-2$^{LRR-1}$, in addition to UBC-3 and UBC-7. One candidate for the latter is the essential LET-70 orthologue of mammalian UBE2D enzymes. A further possibility was suggested by the ~50% synthetic lethality produced by RNAi to *ubxn-3* plus E2 group 6 (Fig 1D), which subsequent deconvolution showed was due to the *ubc-18* orthologue of human UBE2L3 (Fig 1I). This suggested a role for an RBR ligase of the ARIADNE family.

## Reconstitution of replisome-dependent CMG ubiquitylation by CUL-2$^{LRR-1}$ and a suite of ubiquitylation and neddylation enzymes

In order to explore the mechanism by which CUL-2$^{LRR-1}$ promotes replisome-dependent ubiquitylation of the *C. elegans* CMG helicase, we set out to reconstitute the reaction with purified proteins, taking

advantage of recent findings in addition to the results of the E2 screen described above. Work with budding yeast (Deegan *et al*, 2020) and *Xenopus laevis* (Low *et al*, 2020) has shown that CMG ubiquitylation is stimulated by release of the helicase from replication fork DNA, likely reflecting the events that normally lead to CMG disassembly and the termination of DNA replication, when a fork reaches a DNA end such as a telomere or a nick in the DNA template strand upon which the helicase tracks. Based on this observation, replisome-dependent ubiquitylation of the yeast CMG helicase was recently reconstituted with purified proteins in the complete absence of DNA (Deegan *et al*, 2020), reflecting the inherently high efficiency of CMG ubiquitylation, which is repressed throughout elongation by the embrace of the helicase with a replication fork.

Encouraged by these studies, we expressed and purified recombinant forms of the *C. elegans* CMG helicase and associated core replisome proteins (Fig 2A). We also purified a range of ubiquitylation enzymes, together with *C. elegans* NED-8 (equivalent to mammalian NEDD8) and the worm orthologues of mammalian neddylation enzymes (Fig 2A, ULA-1_RFL-1, UBC-12 and DCN-1). Both ubiquitylation and neddylation require an E1 enzyme to activate ubiquitin/NEDD8, which is then transferred to the catalytic cysteine residue of an E2 enzyme (Rennie *et al*, 2020). Subsequently, E3 enzymes mediate the transfer of ubiquitin or NEDD8 from activated E2 to substrate lysine residues (Morreale & Walden, 2016; Zheng & Shabek, 2017), either by bringing E2 and substrate into close proximity and stabilising the active conformation of the E2-Ub or E2-NEDD8 conjugate (e.g. RING E3 ligases including the cullin family), or by transfer of ubiquitin onto a cysteine residue of the E3 and thereafter onto a proximal substrate lysine (e.g. HECT or RBR E3 ligases).

Guided by the analysis of *C. elegans* E2 enzymes described above, we compared the ability of UBC-3, UBC-18 and LET-70 to support the ubiquitylation of recombinant *C. elegans* CMG in reconstituted *in vitro* reactions, which also contained E1, CUL-2$^{LRR-1}$, the worm neddylation machinery and other replisome factors (Fig 2A). In reactions containing UBC-3 as the only E2 enzyme, ubiquitylation of CMG-MCM-7 was not observed (Fig 2B, compare lanes 1 and 5). In contrast, UBC-18 together with the RBR ligase ARI-1 supported the addition of 1–3 ubiquitins to MCM-7 (Fig 2B lane 2; Fig EV1B lanes 9–12 show that both UBC-18 and ARI-1 were required for MCM-7 ubiquitylation). This represented mono-ubiquitylation of multiple sites on MCM-7, since the same ubiquitylation pattern was observed with lysine-free ubiquitin (Fig EV1A, compare lanes 2–3). LET-70 also supported CMG ubiquitylation, with up to ~8 ubiquitins being conjugated to MCM-7 (Fig 2B, lane 3). This predominantly represented the conjugation by LET-70 of a single ubiquitin chain on CMG-MCM-7, since mono-ubiquitylation was the major product in reactions containing LET-70 and lysine-free ubiquitin (Fig EV1A,

**Figure 2. *In vitro* reconstitution of replisome-specific ubiquitylation of CMG-MCM-7 by CUL-2$^{LRR-1}$ and an array of associated ubiquitylation enzymes.**

A  Purified *C. elegans* proteins (see Materials and Methods).
B  Ubiquitylation of CMG-MCM-7 was reconstituted with the indicated factors as described in Materials and Methods. "Neddylation" indicates addition of the *C. elegans* ULA-1_RFL-1 E1 enzyme, the UBC-12 E2 enzyme, the DCN-1 E3 enzyme and NED-8.
C  Equivalent reactions to those in (B) but using lysine-free (K0) ubiquitin.
D  Similar reactions comparing wild-type ubiquitin to the indicated ubiquitin mutants.
E  Reactions containing CMG and either CUL-2$^{LRR-1}$ or CUL-2$^{VHL-1}$ were performed in the presence or absence of other replisome factors as indicated. CMG was then isolated by immunoprecipitation of SLD-5 and the indicated factors were monitored by immunoblotting.

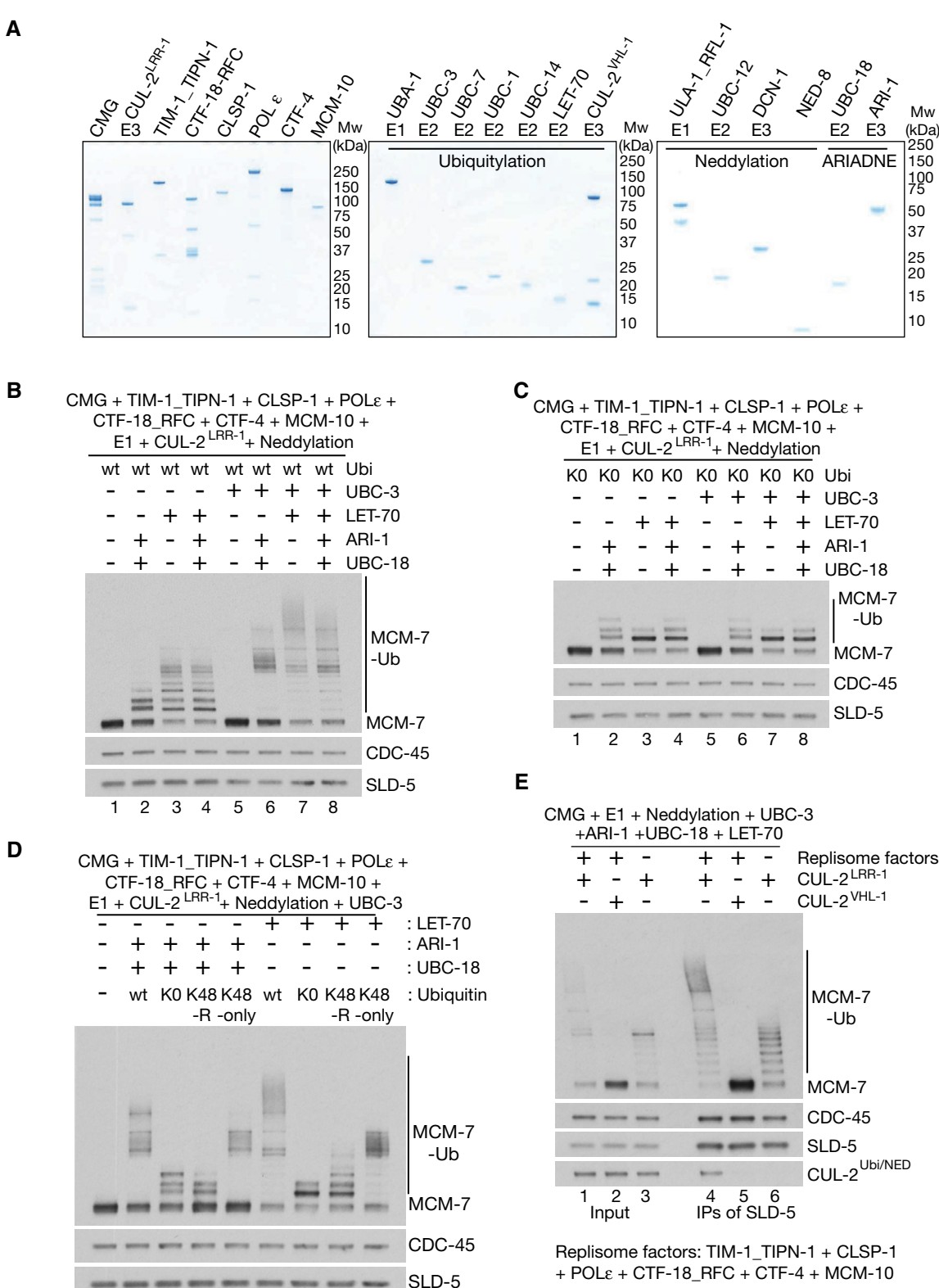

**Figure 2.**

compare lanes 6–7). Moreover, the chains formed by LET-70 were largely independent of lysine 48 of ubiquitin (Fig EV1A, compare lanes 6 and 8). These findings indicated that UBC-18_ARI-1 and LET-70 are both able to prime ubiquitylation of CMG-MCM-7, but cannot synthesise the K48-linked ubiquitin chains that are the preferred substrate of p97-UFD1-NPL4 (Bodnar & Rapoport, 2017, Tsuchiya et al, 2017, van den Boom et al, 2016).

We then performed similar reactions containing CUL-2$^{LRR-1}$ with combinations of UBC-3, LET-70 and UBC-18_ARI-1. These experiments showed that UBC-3 is able to synthesise long ubiquitin chains on CMG-MCM-7, dependent upon the chains having been initiated by either LET-70 or UBC-18_ARI-1 (Fig 2B, lanes 6–7). MCM-7 ubiquitylation in the presence of UBC-3 predominantly takes the form of K48-linked ubiquitin chains (Fig 2D). Moreover, the major product in reactions containing all three E2 enzymes, together with the E3 ligases CUL-2$^{LRR-1}$ and ARI-1, is a single polyubiquitin chain on MCM-7, since reactions containing lysine-free ubiquitin only supported the conjugation of a single ubiquitin moiety to most MCM-7 molecules (compare Fig 2B lane 8 with Fig 2C lane 8). Therefore, these data indicate that LET-70 is the predominant priming enzyme under these reaction conditions (Fig 2C, compare lane 3 with lanes 6–8), whereas UBC-3 is the predominant E2 that extends K48-linked ubiquitin chains on CMG-MCM-7. Consistent with the *in vivo* results of the RNAi E2 screen, we observed in similar *in vitro* reactions that UBC-7 was also able to elongate K48-linked ubiquitin chains that had been primed by LET-70, although UBC-7 was less efficient than UBC-3 (Fig EV1C). In contrast, the more distantly related E2 enzymes UBC-1 and UBC-14 were unable to promote CMG-MCM-7 ubiquitylation (Fig EV1C), as predicted by the RNAi E2 screen (Fig 1C–E).

CMG-MCM-7 ubiquitylation by both UBC-18_ARI-1 and UBC-3 was stimulated by neddylation of conserved lysine residues on CUL-2 (Fig EV1D–F), consistent with previous studies of human CUL-2 (Bandau et al, 2012; Sonneville et al, 2017). This contrasts with the action of yeast SCF$^{Dia2}$ for which neddylation is dispensable (Mukherjee & Labib, 2019; Deegan et al, 2020). Interestingly, CMG-MCM-7 ubiquitylation by *C. elegans* LET-70 did not require neddylation (Fig EV1B, compare lanes 1–4). We found that this was because LET-70 is able to ubiquitylate the neddylation sites on CUL-2 (Fig EV1B lanes 1–2), indicating that cullin ubiquitylation can functionally substitute for cullin neddylation. Correspondingly, mutation of K719 and K749 of CUL-2 abrogated the ubiquitylation or neddylation of CUL-2 and also blocked CMG ubiquitylation (Fig EV1D–F; Fig EV1E shows that CUL-2-2R$^{LRR-1}$ and wild-type CUL-2$^{LRR-1}$ are equally active at promoting free chain formation by UBC-3 in the absence of neddylation).

To assess the importance of the LRR-1 substrate adaptor in the reconstituted CMG ubiquitylation system, we expressed and purified recombinant CUL-2$^{VHL-1}$, in which the worm orthologue of the human Von Hippel-Lindau tumour suppressor (VHL) replaces LRR-1 in the five-subunit E3 ligase (Lisztwan et al, 1999). Both CUL-2$^{VHL-1}$ and CUL-2$^{LRR-1}$ were able to stimulate the formation of free ubiquitin chains by UBC-3 (Fig EV1E). However, only CUL-2$^{LRR-1}$ supported the ubiquitylation of CMG-MCM-7 (Fig 2E, compare lanes 4–5). These data show that the ability of LET-70, UBC-18_ARI-1 and UBC-3 to ubiquitylate CMG-MCM-7 is completely dependent upon the LRR-1 substrate targeting component of CUL-2$^{LRR-1}$.

By isolating the CMG helicase at the end of the reconstituted ubiquitylation reactions, we found that CUL-2$^{LRR-1}$ but not CUL-2$^{VHL-1}$ co-purified with the worm CMG helicase (Fig 2E, CUL-2, lanes 4–5). However, the association of CUL-2$^{LRR-1}$ with CMG was abrogated in reactions that lacked other replisome components except CMG (Fig 2E, CUL-2, compare lanes 4 and 6). Moreover, the efficiency of CMG-MCM-7 ubiquitylation was impaired under such conditions (Fig 2E, MCM-7, compare lanes 4 and 6). These findings indicated that the association of CUL-2$^{LRR-1}$ with the worm CMG helicase is stabilised in the context of the replisome, thereby promoting the efficient ubiquitylation of the CMG-MCM-7 subunit. Nevertheless, residual CMG-MCM-7 ubiquitylation in the absence of other replisome components is still dependent upon CUL-2$^{LRR-1}$ (Fig 2E, MCM-7, compare lanes 5 and 6), likely reflecting a dynamic interaction between LRR-1 and CMG under such conditions.

### TIMELESS-TIPIN links CUL-2$^{LRR-1}$ to the replisome and promotes efficient priming and extension of ubiquitin chains on CMG-MCM-7

The major partners of CMG within the replisome have been best defined in budding yeast (Baretic et al, 2020) and comprise Tof1-Csm3 (TIMELESS-TIPIN in mammals; TIM-1_TIPN-1 in *C. elegans*), Ctf4 (CTF-4/AND-1/WDHD1 in mammals, CTF-4 in *C. elegans*) and Mrc1 (CLASPIN in mammals; CLSP-1 in *C. elegans*). Using purified proteins (Fig 2A), we found that the worm equivalents of these factors also interact with the CMG helicase (Fig EV2A), consistent with the co-purification of such factors with CMG from extracts of *C. elegans* early embryos (Sonneville et al, 2017). To test the role of these factors in promoting the high efficiency of CMG ubiquitylation in the reconstituted ubiquitylation system, we compared the effects of omitting TIM-1_TIPN-1, CTF-4 or CLSP-1. Strikingly, reactions omitting TIM-1_TIPN-1 had an equivalent defect to reactions that lacked all replisome components apart from CMG (Fig 3A, MCM-7, compare lanes 2–3). In contrast, removal of CTF-4 or CLSP-1 had no impact on the efficiency of CMG-MCM-7 ubiquitylation (Fig 3A, MCM-7, compare lane 1 with lanes 4–5). These findings contrasted with studies of CMG ubiquitylation in budding yeast (Maculins et al, 2015; Deegan et al, 2020), in which Ctf4 and Mrc1/CLASPIN were found to be crucial for action of the cullin 1 ligase SCF$^{Dia2}$ during DNA replication termination.

Subsequently, titration experiments indicated that the ability of CUL-2$^{LRR-1}$ to mediate the priming of ubiquitin chains on CMG-MCM-7 was stimulated several-fold in the presence of TIM-1_TIPN-1 (Figs 3B and Fig EV2B and C, K0 = lysine-free ubiquitin). Moreover, TIM-1_TIPN-1 stimulated priming by both ARI-1_UBC-18 (Fig EV2D lower panels) and LET-70 (Fig EV2E lower panels). Similarly, the elongation of K48-linked ubiquitin chains on CMG-MCM-7 was stimulated in the presence of TIM-1_TIPN-1 (Figs 3C and EV2D and E upper panels). These data suggested that TIM-1_TIPN-1 functions by stimulating CUL-2$^{LRR-1}$ function, which in turn is required for all aspects of CMG-MCM-7 ubiquitylation.

To test whether the TIM-1_TIPN-1 complex is required for stable association of CUL-2$^{LRR-1}$ with the *C. elegans* replisome, we isolated the CMG helicase from a mixture of CUL-2$^{LRR-1}$ and replisome proteins and observed that the co-purification of the cullin ligase with CMG was dependent upon TIM-1_TIPN-1 but not CTF-4 or CLSP-1 (Fig 3D, CUL-2). Moreover, glycerol gradient analysis of

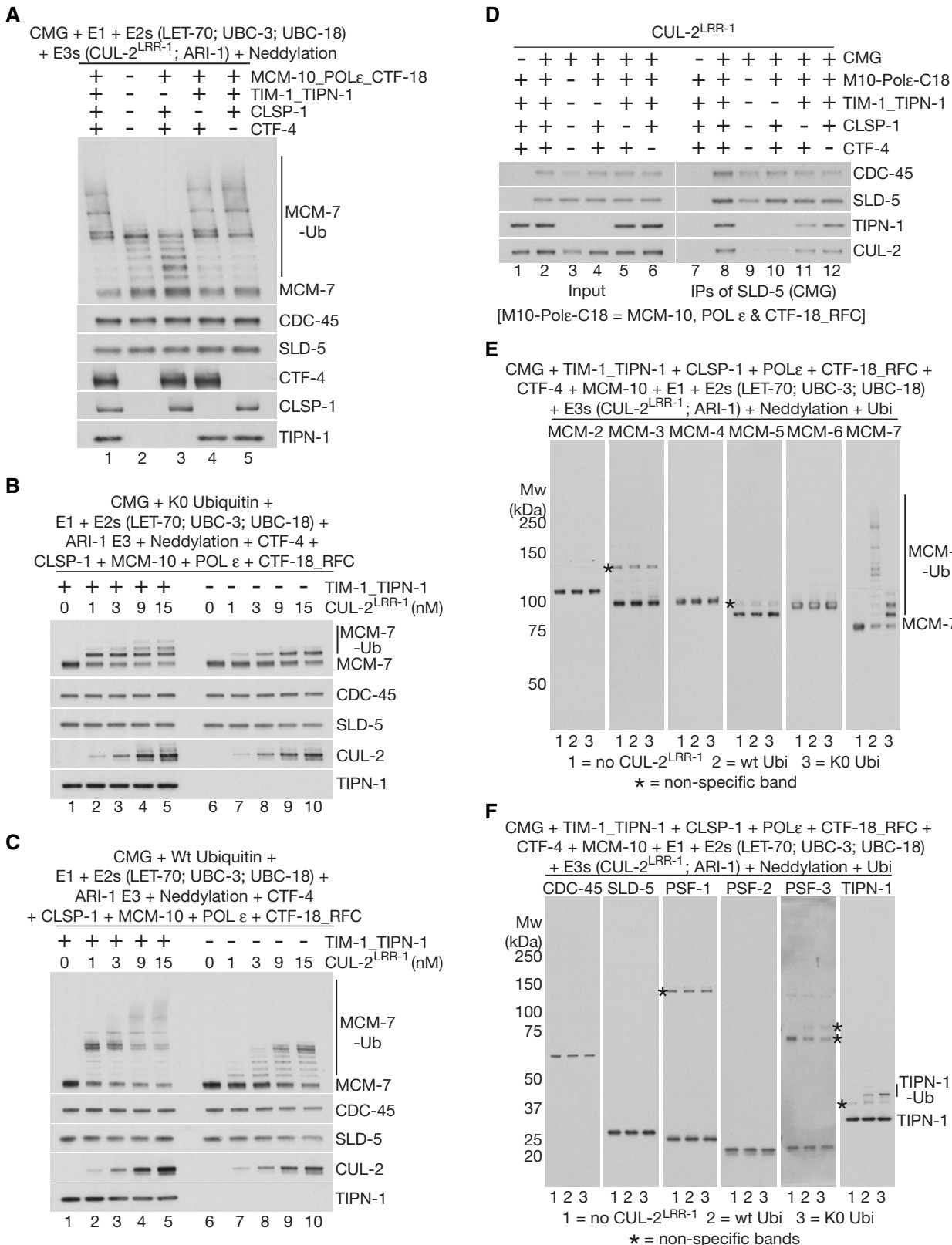

**Figure 3.**

◀

**Figure 3.  TIMELESS-TIPIN stimulates the priming and elongation of ubiquitin chains on CMG-MCM-7 by stabilising the association of CUL-2$^{LRR-1}$ with the *C. elegans* replisome.**

A–C  Reconstituted CMG-MCM-7 ubiquitylation reactions were performed as above in the presence of the indicated factors. "Neddylation" indicates addition of the *C. elegans* ULA-1_RFL-1 E1 enzyme, the UBC-12 E2 enzyme, the DCN-1 E3 enzyme and NED-8.
D  The indicated factors were assembled in the absence of ubiquitin and then incubated as for (A) before isolation of CMG via immunoprecipitation of SLD-5. The association of CUL-2 with the CMG-replisome was monitored by immunoblotting.
E, F  In reactions analogous to those in (A), ubiquitylation of the indicated factors was monitored by immunoblotting. Reactions were performed in sets of three as indicated (1 = dropout of CUL-2$^{LRR-1}$; 2 = wt ubiquitin; 3 = lysine-free or K0 ubiquitin).

mixtures of purified CMG, TIM-1_TIPN-1 and CUL-2$^{LRR-1}$ indicated that the co-migration of CUL-2$^{LRR-1}$ with either CMG or TIM-1_TIPN-1 was dependent upon the presence of all three factors (Fig EV2F). These data likely reflect a direct interaction of CUL-2$^{LRR-1}$ with both CMG and TIM-1_TIPN-1, which normally occurs in the context of the replisome. Consistent with the notion that CUL-2$^{LRR-1}$ interacts with both CMG and TIM-1_TIPN-1, we found that although MCM-7 was the preferred substrate of CUL-2$^{LRR-1}$ amongst the 11 subunits of CMG and a range of other replisome factors, both subunits of TIM-1_TIPN-1 were also detectably ubiquitylated at a low level (Figs 3E and F, and EV2G). Moreover, the ubiquitylation of the TIM-1_TIPN-1 complex was detectable in the absence of CMG and other replisome factors, further indicating a direct interaction between CUL-2$^{LRR-1}$ and TIM-1_TIPN-1. Nevertheless, ubiquitylation of TIM-1_TIPN-1 was further stimulated in the presence of CMG (Fig EV2H; note that the presence of other replisome factors did not affect ubiquitylation of TIM-1_TIPN-1), probably aided by the formation of a ternary complex between the ligase, helicase and TIM-1_TIPN-1 (as described above and shown in Fig EV2F).

## TIMELESS-TIPIN drives CMG over the "ubiquitin threshold" for disassembly by CDC-48

To explore the functional significance of enhanced CMG ubiquitylation by CUL-2$^{LRR-1}$ in the presence of TIM-1_TIPN-1, we reconstituted disassembly of the ubiquitylated helicase with purified proteins. Ubiquitylation reactions were performed in the presence of TIMELESS-TIPIN and other replisome factors, before isolating ubiquitylated CMG on beads that were coated with antibodies to the GINS subunit SLD-5. The ubiquitylated helicase was then incubated with purified *C. elegans* CDC-48, UFD-1_NPL-4 and UBXN-3 (Fig 4A and B). Successful CMG disassembly was indicated by CDC-48 dependent release of CDC-45 and MCM-2-7 into the supernatant, whilst GINS remained bound to antibody on the beads (Fig 4C, compare lanes 3 and 4). Notably, MCM-7 with less than ~five attached ubiquitins was retained on the beads, indicating that helicase disassembly was dependent upon the conjugation of around five or more ubiquitin moieties to the helicase (Fig 4C, compare lanes 3–4). These findings indicate that metazoan CDC-48_UFD-1_NPL-4 is governed by an analogous "ubiquitin threshold" to its yeast counterpart (Bodnar & Rapoport, 2017; Twomey *et al*, 2019; Deegan *et al*, 2020).

Importantly, the proportion of CMG-MCM-7 above this ubiquitin threshold was much reduced and CMG disassembly was correspondingly less efficient, when reactions were performed in the absence of TIM-1_TIPN-1 (Fig 4D, compare lanes 3–4). Together with the data presented above, these findings indicate that the TIMELESS-TIPIN complex promotes efficient CMG disassembly, by

stimulating the priming and elongation of K48-linked ubiquitin chains on CMG-MCM-7.

## UBXN-3 stimulates disassembly of ubiquitylated CMG by CDC-48_UFD-1_NPL-4

The reconstituted CMG disassembly reaction was dependent upon the UFD-1_NPL-4 complex (Fig 4C, compare lanes 3 and 5), which is predicted by studies of its yeast counterpart to recognise K48-linked ubiquitin chains and stimulate the initial unfolding of a substrate-linked ubiquitin moiety by CDC-48 (Bodnar & Rapoport, 2017, Twomey *et al*, 2019, van den Boom *et al*, 2016). The mechanism of substrate unfolding by CDC-48/p97 in association with UFD1-NPL4 has not previously been shown to involve any additional factors. Strikingly, however, the disassembly of ubiquitylated CMG was very inefficient in reactions that lacked UBXN-3 and only contained CDC-48 and UFD-1_NPL-4 (Fig 4C, compare lanes 3-4 with lanes 7-8). Moreover, the same was true in reactions where *C. elegans* CDC-48 and adaptors were used to disassemble ubiquitylated yeast CMG helicase (R. Fujisawa and K. Labib, unpublished data), indicating that the importance of UBXN-3 was not dependent upon specific interactions with components of the worm replisome.

The stimulation of CMG disassembly was dependent upon the UBX domain of UBXN-3 (Fig EV3, compare lanes 5 and 7), which interacts directly with CDC-48 (Franz *et al*, 2016). In addition, CMG disassembly by CDC-48_UFD-1_NPL-4 and UBXN-3 was dependent upon the presence of a K48-linked ubiquitin chain on CMG, since disassembly did not occur in reactions containing lysine-free ubiquitin (Fig EV3, lane 9) or K48R ubiquitin (Fig EV3, lane 11). Overall, these data indicate that UBXN-3 directly stimulates the disassembly of poly-ubiquitylated CMG by *C. elegans* CDC-48_UFD-1_NPL-4. Consistent with UBXN-3 playing an important role in the biology of *C. elegans* CDC-48_UFD-1_NPL-4, we found that fertility and embryonic viability were extremely low upon deletion of the *ubxn-3* gene by CRISPR-Cas9 (Appendix Fig S1E–H). RNAi depletion of UBXN-3 is not lethal (Sasagawa *et al*, 2010; Sonneville *et al*, 2017), presumably due to the presence of residual protein. Nevertheless, previous work showed that *ubxn-3* RNAi causes synthetic lethality in worms with reduced expression of CDC-48 (Franz *et al*, 2016), further indicating the importance of UBXN-3 for CDC-48 biology in *C. elegans*.

## TIMELESS-TIPIN is important for efficient CMG ubiquitylation during DNA replication termination in *Caenorhabditis elegans* early embryos

To begin to investigate the *in vivo* significance of the reconstituted CMG ubiquitylation and disassembly reactions, for CMG ubiquitylation and replisome disassembly during DNA replication termination,

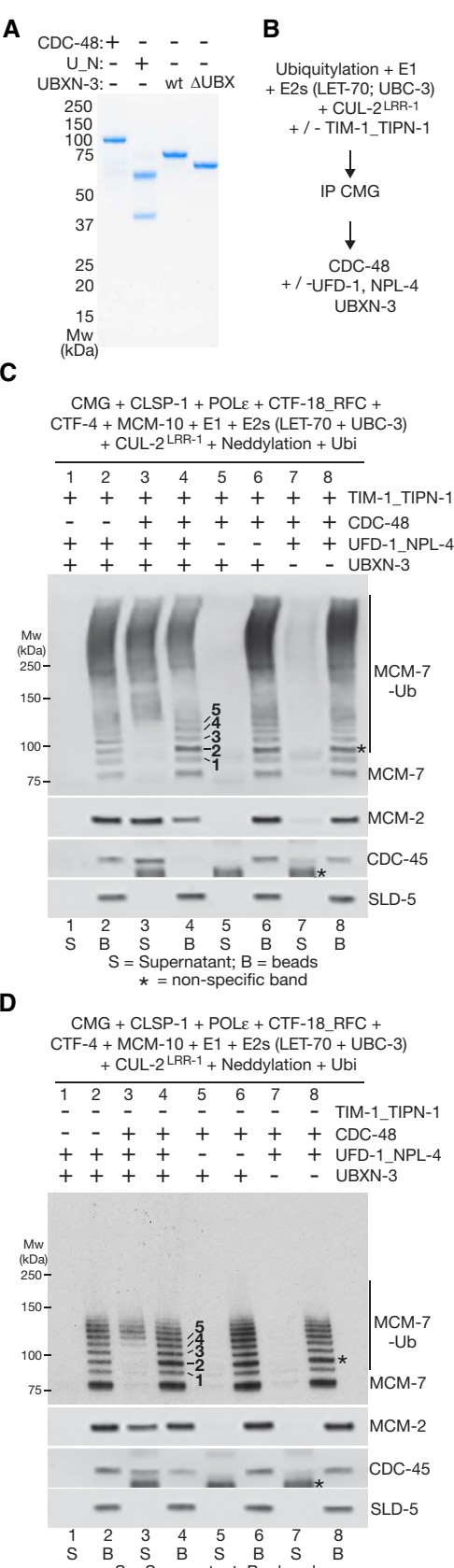

**A** 
CDC-48: + − − − 
U_N: − + − − 
UBXN-3: − − wt ΔUBX 

Mw (kDa): 250, 150, 100, 75, 50, 37, 25, 20, 15

**B** 
Ubiquitylation + E1 
+ E2s (LET-70; UBC-3) 
+ CUL-2^LRR-1 
+ / − TIM-1_TIPN-1 

↓ 

IP CMG 

↓ 

CDC-48 
+ / −UFD-1, NPL-4 
UBXN-3

**C** 
CMG + CLSP-1 + POLε + CTF-18_RFC + 
CTF-4 + MCM-10 + E1 + E2s (LET-70 + UBC-3) 
+ CUL-2^LRR-1 + Neddylation + Ubi

| 1 | 2 | 3 | 4 | 5 | 6 | 7 | 8 | |
|---|---|---|---|---|---|---|---|---|
| + | + | + | + | + | + | + | + | TIM-1_TIPN-1 |
| − | − | + | + | + | + | + | + | CDC-48 |
| + | + | + | + | − | − | + | + | UFD-1_NPL-4 |
| + | + | + | + | + | + | − | − | UBXN-3 |

Mw (kDa): 250, 150, 100, 75

MCM-7-Ub, MCM-7, MCM-2, CDC-45, SLD-5

5, 4, 3, 2, 1

1 2 3 4 5 6 7 8 
S B S B S B S B 
S = Supernatant; B = beads 
★ = non-specific band

**D** 
CMG + CLSP-1 + POLε + CTF-18_RFC + 
CTF-4 + MCM-10 + E1 + E2s (LET-70 + UBC-3) 
+ CUL-2^LRR-1 + Neddylation + Ubi

| 1 | 2 | 3 | 4 | 5 | 6 | 7 | 8 | |
|---|---|---|---|---|---|---|---|---|
| − | − | − | − | − | − | − | − | TIM-1_TIPN-1 |
| − | − | + | + | + | + | + | + | CDC-48 |
| + | + | + | + | − | − | + | + | UFD-1_NPL-4 |
| + | + | + | + | + | + | − | − | UBXN-3 |

Mw (kDa): 250, 150, 100, 75

MCM-7-Ub, MCM-7, MCM-2, CDC-45, SLD-5

5, 4, 3, 2, 1

1 2 3 4 5 6 7 8 
S B S B S B S B 
S = Supernatant; B = beads 
★ = non-specific band

**Figure 4. UBXN-3 stimulates disassembly of ubiquitylated CMG by CDC-48_UFD-1_NPL-4.**

A CDC-48, UFD-1_NPL-4 (U_N) and UBXN-3 (full-length and UBXN-3-ΔUBX) were purified as described in Materials and Methods and analysed by Coomassie blue staining of SDS–PAGE gels.

B Ubiquitylation and disassembly of the *C. elegans* CMG helicase was reconstituted *in vitro* as indicated (see Materials and Methods for further details).

C CMG was isolated by immunoprecipitation of SLD-5, after ubiquitylation in the presence of TIM-1_TIPN-1, before incubation with the indicated factors. "Neddylation" indicates addition of the *C. elegans* ULA-1_RFL-1 E1 enzyme, the UBC-12 E2 enzyme, the DCN-1 E3 enzyme and NED-8. CMG disassembly was monitored by release of CDC-45 or MCM-2-7 proteins into the supernatant, whilst SLD-5 and other GINS subunits remained on the beads. Asterisks indicate non-specific bands in the anti-MCM-7 and CDC-45 immunoblots.

D Equivalent reactions to (C) involving ubiquitylation of CMG in the absence of TIM-1_TIPN-1. For (C) and (D), the bands corresponding to MCM-7 conjugated to 1–5 ubiquitins are indicated in lane 4.

we used RNAi to deplete the major CMG partner proteins in the *C. elegans* early embryo. These experiments used worms in which the PSF-1 subunit of GINS was modified either with the tandem affinity purification (TAP) tag to allow for immunoprecipitation of CMG, or with green fluorescent protein (GFP) to monitor the presence of CMG on condensed chromatin during mitosis.

We first examined the importance of core replisome proteins for CMG-MCM-7 ubiquitylation during DNA replication termination. Since ubiquitylated CMG is disassembled very rapidly during DNA replication termination by CDC-48_UFD-1_NPL-4 (Sonneville *et al*, 2017), we co-depleted NPL-4 along with the CMG partner proteins. Depletion of NPL-4 in *TAP-psf-1* worms led to the accumulation of post-termination CMG helicase with a mixture of ubiquitylated and non-ubiquitylated MCM-7 subunit (Fig 5A, lane 5; the fraction of non-ubiquitylated CMG likely results from partial depletion of the pool of free ubiquitin upon inactivation of CDC-48_UFD-1_NPL-4).

CMG-MCM-7 ubiquitylation in worms lacking NPL-4 was unaffected by additional depletion of either CTF-4 or CLSP-1 (Fig 5A; compare lanes 5-7). To exclude the potential contribution of residual protein after RNA interference, we deleted the *ctf-4* gene by CRISPR-Cas9 (Fig EV4A and B and Materials and Methods; note that deletion of *clsp-1* is lethal in *C. elegans*, precluding a similar approach). Upon exposure to *npl-4* RNAi, the accumulation of ubiquitylated CMG helicase and the replisome association of CUL-2^LRR-1 was comparable in control and *ctf-4Δ* worms (Fig EV4C, compare lanes 5 and 7), even upon additional RNAi depletion of CLSP-1 (Fig EV4C, compare lanes 7 and 8). Therefore, CTF-4 and CLSP-1 are dispensable for the efficiency of CMG-MCM-7 ubiquitylation in the *C. elegans* early embryo.

In contrast, exposing *TAP-psf-1* worms to *tim-1* RNAi produced a dramatic reduction in CMG-MCM-7 ubiquitylation and abrogated the association of CUL-2^LRR-1 with the replisome (Fig 5A and B). Moreover, depletion of TIPN-1 produced a similar though slightly milder defect compared to *tim-1* RNAi (Fig 5B, *tipn-1* RNAi, lanes 4 + 8). For multiple reasons, the effect on CMG ubiquitylation of depleting TIMELESS-TIPIN is unlikely to be due to arrested progression of DNA replication forks, such as produced by RNAi inactivation of ribonucleotide reductase or DNA polymerase alpha (Sonneville *et al*, 2017). Firstly, *tim-1* RNAi only had a mild impact

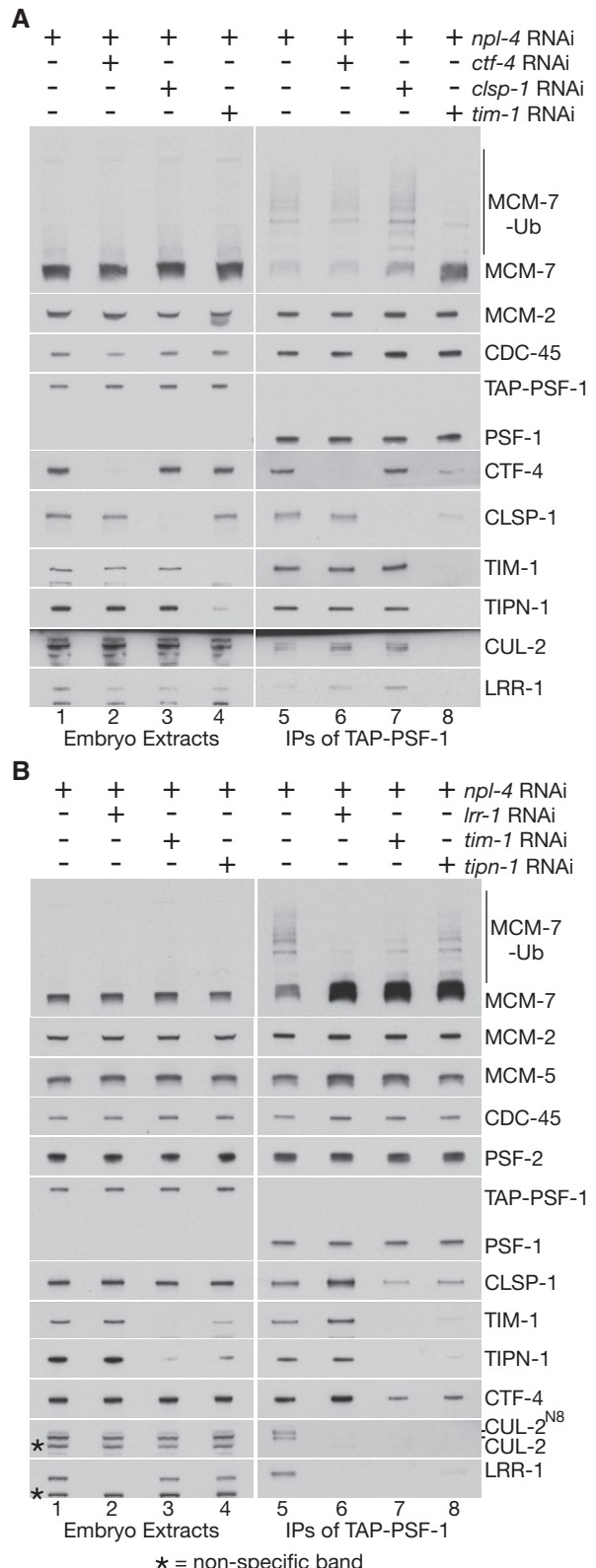

**Figure 5. The TIMELESS-TIPIN complex is required in *C. elegans* early embryos for efficient CMG helicase ubiquitylation and the association of CUL-2^LRR-1 with the post-termination replisome.**

A  *TAP-psf-1* worms were fed on bacteria containing a single plasmid expressing the indicated RNAi treatments, before preparation of embryonic cell extracts and isolation of TAP-PSF-1 by immunoprecipitation. The bound material was released from beads by cleavage of the TAP tag with TEV protease, before detection of the indicated factors by immunoblotting.

B  Similar experiment comparing the impact of RNAi inactivation of *lrr-1*, *tim-1* and *tipn-1* upon CMG-MCM-7 ubiquitylation and the association of CUL-2^LRR-1 with the worm replisome. Asterisks indicate non-specific bands in the immunoblots.

impair chromatin condensation during mitosis (see Fig 6B below) and did not cause the single-strand DNA binding protein RPA to accumulate on chromatin (Appendix Fig S3). Instead, these findings support the *in vitro* reconstitution data described above (Fig 3) and indicate that TIMELESS-TIPIN is important *in vivo* for the normally high efficiency of CMG helicase ubiquitylation during DNA replication termination in the *C. elegans* early embryo.

**Co-depletion of TIM-1 and UBXN-3 causes synthetic CMG disassembly defects during DNA replication termination and loss of embryonic viability**

In the reconstituted CMG disassembly system described above, removal of either TIMELESS-TIPIN or UBXN-3 produced defects in CMG helicase disassembly (Fig 4C and D), reflecting the respective contributions of TIMELESS-TIPIN to CMG ubiquitylation and UBXN-3 to CMG disassembly. To test whether combined RNAi inactivation of TIMELESS-TIPIN and UBXN-3 *in vivo* would produce an additive defect in CMG helicase chromatin extraction, during DNA replication termination in the *C. elegans* early embryo, we used spinning disc confocal microscopy to assay for persistence of a CMG subunit (GFP-tagged PSF-1) on condensed mitotic chromatin. As reported previously, RNAi depletion of *lrr-1* was used a positive control for defective CMG disassembly during DNA replication termination (Sonneville *et al*, 2017).

Individual depletion of either TIM-1 or UBXN-3 did not lead to detectable accumulation of GFP-PSF-1 on chromatin during mitotic prophase (Fig 6B, *tim-1* RNAi or *ubxn-3* RNAi), in contrast to RNAi inactivation of *lrr-1* (Fig 6B, *lrr-1* RNAi). This indicated that any defect in CMG disassembly upon RNAi depletion of either TIM-1 or UBXN-3 was partial or transient. However, GFP-PSF-1 persisted on chromatin throughout mitosis after co-depletion of TIM-1 and UBXN-3 (Fig 6B, *tim-1 ubxn-3* RNAi), as seen upon co-depletion of LRR-1 and UBXN-3 (Fig 6B, *lrr-1 ubxn-3* RNAi). The presence of GFP-PSF-1 on chromatin during prophase under such conditions indicated that the combined absence of TIM-1 and UBXN-3 produced a synthetic defect in CMG disassembly during DNA replication termination. Subsequently, CMG then persisted on chromatin throughout mitosis, due to the additional role of UBXN-3 in the mitotic CMG disassembly pathway, which also requires TRUL-1 and CDC-48_UFD-1_NPL-4 (Sonneville *et al*, 2017; Sonneville *et al*, 2019). Notably, *tim-1* RNAi did not lead to the persistence of GFP-PSF-1 on mitotic chromatin in *trul-1Δ* worms that lacked the mitotic CMG disassembly pathway (Fig EV5A and B). These findings indicated that UBXN-3 is important for chromatin extraction of CMG

on embryonic viability (Fig 6A) and did not reduce the total amount of the CMG helicase in early embryos (Fig 5A, lanes 5-8, compare PSF-1, CDC-45 and MCM-2). Furthermore, *tim-1* RNAi did not

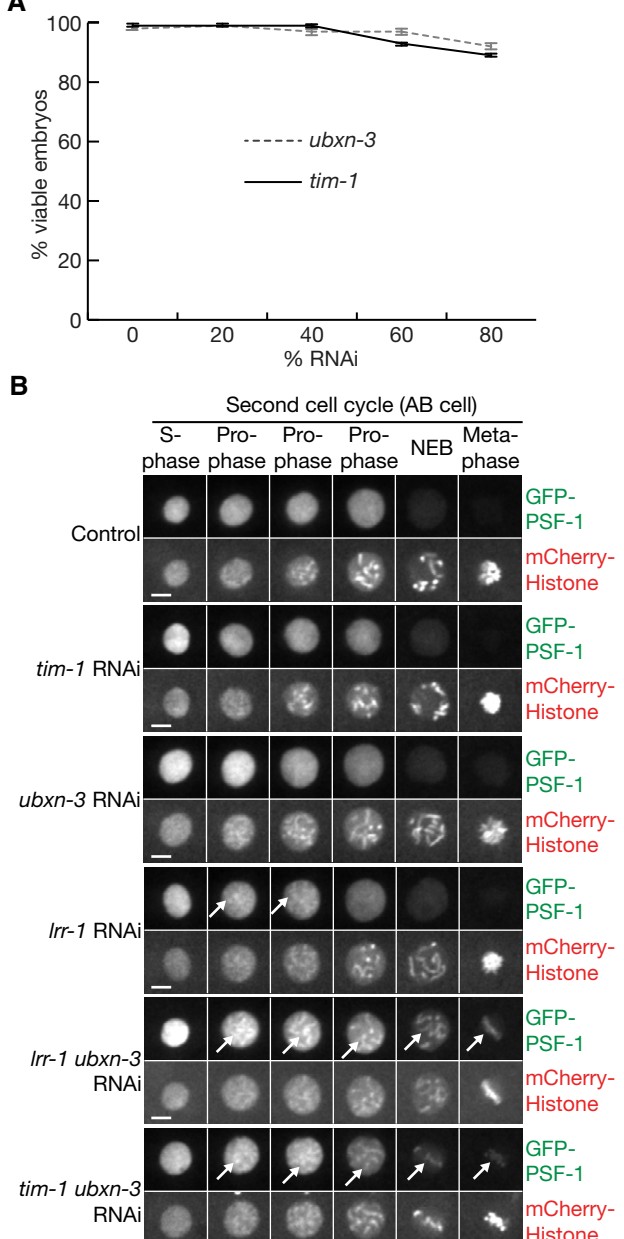

A  Embryonic viability was monitored as described in Materials and Methods, after feeding worms on the indicated proportions of bacteria expressing RNAi ("% RNAi") to the denoted genes, mixed with bacteria containing empty vector (to make a total of 100%).

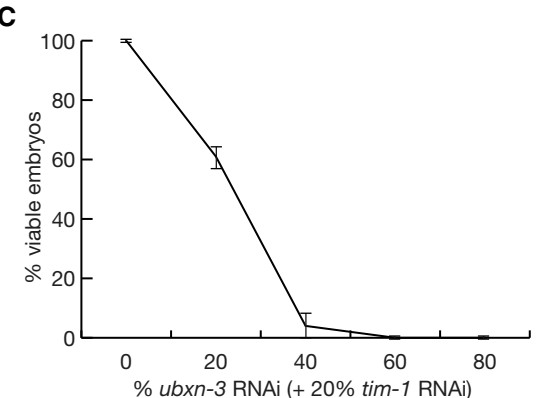

**Figure 6. Co-depletion of TIMELESS and UBXN-3 impairs CMG disassembly during DNA replication termination in *C. elegans* and causes loss of viability.**

A  Embryonic viability was monitored as described in Materials and Methods, after feeding worms on the indicated proportions of bacteria expressing RNAi ("% RNAi") to the denoted genes, mixed with bacteria containing empty vector (to make a total of 100%).

B  The presence of GFP-PSF-1 on mitotic chromatin (indicated by white arrows) was monitored by spinning disc confocal microscopy (see Materials and Methods), in embryos derived from worms that were fed on bacteria containing a single plasmid expressing the indicated RNAi ("Control" = empty vector). The scale bars correspond to 5 μm.

C  Analogous experiment to that in (A), but with worms fed on the indicated proportions of bacteria expressing *ubxn-3* RNAi, in combination with 20% bacteria expressing *tim-1* RNAi specific for *tim-1*, plus the required remainder of bacteria containing empty vector (to make a total of 100%).

Data information: The data in (A) and (C) represent the means and standard deviations from three biological replicates.

during DNA replication termination in worms depleted for TIM-1, distinct from the previously characterised role of UBXN-3 during mitosis (Sonneville *et al*, 2017).

Since co-depletion of TIM-1 and UBXN-3 produced a synthetic defect in the CUL-2$^{\text{LRR-1}}$ pathway of CMG disassembly during DNA replication termination, and LRR-1 is essential for the production of viable offspring in *C. elegans* (Merlet *et al*, 2010), we investigated the impact on embryonic viability of combined RNAi to *tim-1* and *ubxn-3*. As discussed above, viability remained high when worms were exposed to increasing doses of *tim-1* RNAi, and the same was true for *ubxn-3* RNAi (Fig 6A). This likely reflected imperfect depletion of the target protein, since CRISPR-Cas9 deletion of *tim-1* is lethal (R. Sonneville, Y. Xia, Y. Hong and K. Labib, unpublished data) and *ubxn-3*Δ also produces a dramatic loss of viability as noted above (Appendix Fig S1E–H).

However, even partial depletion of TIM-1 produced a complete loss of viability in combination with partial depletion of UBXN-3 (Fig 6C), analogous to the effects of RNAi co-depletion of UBC-3, UBC-7 and UBXN-3 (Fig 1E and F). In contrast, RNAi depletion of TIM-1 or UBC-3+UBC-7, or partial RNAi depletion of LRR-1, did not cause an equivalent loss of viability in *trul-1*Δ worms that lack the mitotic pathway for CMG disassembly (Fig EV5C). These findings indicated that the strong synthetic lethal phenotype that is produced by co-depletion of TIM-1 and UBXN-3 likely reflects the associated defect in CMG helicase disassembly during DNA replication termination.

## Discussion

Through a combination of *in vitro* reconstitution and *in vivo* analysis in the *C. elegans* early embryo, we have identified an important role for the TIMELESS-TIPIN complex in the CUL-2$^{\text{LRR-1}}$ pathway for CMG helicase ubiquitylation during DNA replication termination. This contrasts with the situation during replication termination in budding yeast, where the efficiency of CMG ubiquitylation is enforced by recruitment of the unrelated E3 ligase SCF$^{\text{Dia2}}$ by the replisome factors Ctf4 and Mrc1.

Our data show that *C. elegans* TIMELESS-TIPIN stimulates CMG ubiquitylation several-fold, by stabilising the association of CUL-

2^LRR-1 with the replisome complex that assembles around the CMG helicase (Fig 3). Cryo-electron microscopy of the yeast replisome has shown that yeast TIMELESS-TIPIN binds to the amino-terminal tier of the Mcm-2-7 component of CMG, as well as to duplex DNA just ahead of the replisome (Baretic *et al*, 2020). As the LRR-1 substrate adaptor of CUL-2^LRR-1 is essential for CMG ubiquitylation (Fig 2E), and TIMELESS-TIPIN itself is ubiquitylated by CUL-2^LRR-1 (Figs 3F and EV2G and H), it is likely that LRR-1 binds directly to TIMELESS-TIPIN in the context of the replisome, as well as binding to components of CMG. We propose that these contacts jointly position the ligase in order to direct efficient ubiquitylation of CMG-MCM-7. Structural biology will play a key role in exploring this model further in future studies.

In both yeast and metazoa, the TIMELESS-TIPIN complex governs multiple aspects of DNA replication fork biology. For example, TIMELESS and TIPIN are important for the rate of fork progression and are required for activation of the S-phase checkpoint pathway that helps cells to survive DNA replication stress (Tourriere *et al*, 2005; Unsal-Kacmaz *et al*, 2007; Smith *et al*, 2009; Westhorpe *et al*, 2020). In addition, the TIMELESS-TIPIN complex recruits additional factors to forks, such as the DDX11 helicase that unwinds G4 motifs that can form on the parental DNA template during the course of replication (Lerner *et al*, 2020). Thus, TIMELESS and TIPIN are important genome stability factors at replication forks. Our work identifies a new mechanism by which the metazoan TIMELESS-TIPIN complex preserves genome integrity, namely by stimulating CMG helicase ubiquitylation and disassembly during DNA replication termination in the *C. elegans* early embryo.

Using purified *C. elegans* proteins to reconstitute the action of metazoan CDC-48/p97 and its adaptor proteins *in vitro*, we found that UBXN-3 greatly stimulates the disassembly of ubiquitylated CMG by CDC-48_UFD-1_NPL-4 (Fig 4). Such a role would not have been anticipated from past studies of yeast or human Cdc48/p97, which showed that the Ufd1-Npl4 adaptor proteins are necessary and sufficient to recognise and unfold a model ubiquitylated substrate based on a poly-ubiquitylated fluorescent protein (Blythe *et al*, 2017; Bodnar & Rapoport, 2017; Twomey *et al*, 2019; Pan *et al*, 2021). Moreover, yeast Cdc48-Ufd1-Npl4 processes ubiquitylated CMG with high efficiency *in vitro*, unfolding the ubiquitylated Mcm-7 subunit and disrupting the integrity of the helicase, without requiring an additional adaptor of Cdc48 (Maric *et al*, 2017; Mukherjee & Labib, 2019; Deegan *et al*, 2020). Worm UBXN-3 co-purifies from worm cell extracts with both CDC-48 and UFD-1_NPL-4 (Sasagawa *et al*, 2010; Franz *et al*, 2016), and studies of recombinant human FAF1 have shown that it can bind directly to p97-UFD1-NPL4 (Hanzelmann *et al*, 2011). In future work, structural approaches will be important to further elucidate the mechanism by which UBXN-3/FAF1 co-operates with UFD1-NPL4 to direct the disassembly of ubiquitylated complexes by CDC-48/p97.

UBXN-3 is critically important *in vivo* for CMG disassembly upon RNAi inactivation of TIMELESS-TIPIN (Fig 6B) that reduces the efficiency of CMG ubiquitylation during DNA replication termination (Fig 5). Correspondingly, even partial co-depletion of TIMELESS and UBXN-3 induces synthetic lethality. These findings are potentially of interest in the context of human cancer, for two main reasons. Firstly, TIMELESS and TIPIN are co-ordinately over-expressed in many human cancer cells (Bianco *et al*, 2019), where they are important to combat the inherent DNA replication stress that is a feature

of the cancerous state. Secondly, the FAF1 human orthologue of *C. elegans* UBXN-3 is a candidate tumour suppressor (Menges *et al*, 2009; Bonjoch *et al*, 2020). It will be important in future work to characterise the p97 adaptors that are required for CMG helicase disassembly in mammalian cells, in addition to UFD1-NPL4. Should mammalian FAF1 play an analogous role during CMG helicase disassembly to *C. elegans* UBXN-3, it would then be interesting to explore whether partial inactivation of TIMELESS-TIPIN, for example via Proteolysis Targeting Chimeras or PROTACs (Maniaci & Ciulli, 2019; Verma *et al*, 2020), might induce synthetic lethality in cancer cells that lose the FAF1 gene during their development.

## Materials and Methods

### Strains and plasmids

The *C. elegans* strains used in this study were derived from the "Bristol N2" wild type and are described in Appendix Table S1. Alleles generated via the CRISPR-Cas9 genome editing system (InVivo Biosystems & Suny Biotech) were subsequently out-crossed eight times with the N2 wild-type *C. elegans* strain. The strain KAL55 was generated by crossing KAL17 with KAL21. KAL92 was made by crossing KAL90 with KAL3.

For expression of proteins in budding yeast, and as detailed in Appendix Tables S1–S3, one of the three *Saccharomyces cerevisiae* strains yJF1 (*MAT**a** ade2-1 ura3-1 his3-11,15 trp1-1 leu2-3,112 can1-100 bar1Δ::hphNT pep4Δ::kanMX*), YSS3 (*MAT**a** ade2-1 ura3-1 his3-11,15 trp1-1 leu2-3,112 can1-100 pep4Δ::ADE2*) or YSS4 (*MATα ade2-1 ura3-1 his3-11,15 trp1-1 leu2-3,112 can1-100 pep4Δ::ADE2*) was transformed with the indicated linearised plasmids using standard procedures. The codon usage of the expression constructs was optimised for high-level expression in *Saccharomyces cerevisiae*, as described previously (Yeeles *et al*, 2015). The codon optimised DNA sequences were synthesised by GenScript.

For expression of proteins in bacteria, the plasmids listed in Appendix Table S1 and S3 were transformed into the *E. coli* strain Rosetta™ (DE3) pLysS (70956, Novagen).

### *Caenorhabditis elegans* maintenance

Worms were maintained according to standard procedures and were grown on "Nematode Growth Medium" (NGM: 3 g/l NaCl; 2.5 g/l peptone; 20 g/l agar; 5 mg/l cholesterol; 1 mM CaCl$_2$; 1 mM MgSO$_4$; 2.7 g/l KH$_2$PO$_4$; 0.89 g/l K$_2$HPO$_4$).

### RNA interference

RNAi was performed by feeding worms with bacteria containing plasmids that express double-stranded RNA. RNAse III-deficient HT115 bacteria were transformed with an indicated L4440-derived plasmid. For microscopy experiments, worms were fed on 6-cm plates containing the following medium: 3 g/l NaCl, 20 g/l agarose, 5 mg/l cholesterol, 1 mM CaCl$_2$, 1 mM MgSO$_4$, 2.7 g/l KH$_2$PO$_4$, 0.89 g/l K$_2$HPO$_4$, 1 mM IPTG and 100 mg/l Ampicillin. For immunoprecipitation experiments, worms were fed on 15 cm plates containing NGM medium supplemented with 1 mM IPTG and 100 mg/l Ampicillin.

The plasmids expressing dsRNA were either derived from a commercial RNAi library (SourceBioscience, UK; *ctf-4, clsp-1* and *pola-1*), or else were made by cloning PCR products into the vector L4440. In the latter case, we either amplified 1 kb products from cDNA (*npl-4.2 isoform a, lrr-1, ubxn-3, tim-1, tipn-1, ubc-3 isoform a, ubc-7, ubc-26, ubc-15 isoform f, ubc-6 isoform a, ubc-22, ubc-25, ubc-23, ubc-13, ubc-21, ubc-20, ubc-17, ubc-16, ubc-14, ubc-1, ubc-9, ubc-12, ubc-8, ubc-19, ubc-18*), or amplified full-length cDNA for open reading frames shorter than 1 kb, using a cDNA library that was kindly provided by Sarah-Lena Offenburger. Details of sequences used in the RNAi vectors are provided in Appendix Table S1.

To target more than one gene simultaneously by RNAi, as indicated above we either fed a mixture of bacteria expressing the corresponding dsRNA or else cloned contiguous 1 kb fragments for each gene into a single L4440 plasmid. Empty L4440 vector was used as the control for RNAi experiments throughout this study.

When screening the set of E2s by RNAi, we generated L4440-derived vectors containing contiguous combinations of DNA fragments to target multiple genes simultaneously in groups (*Group 1: ubc-26 + ubc-15 isoform f + ubc-6 isoform a; Group 2: ubc-22 + ubc-25 + ubc-23; Group 3: ubc-13 + ubc-21 + ubc-20; Group 4: ubc-17 + ubc-16; Group 5: ubc-14 + ubc-3 isoform a + ubc-7 + ubc-1; Group 6: ubc-8 + ubc-19 + ubc-18*).

## Microscopy

Worms at the larval L4 stage were incubated on 6-cm RNAi feeding plates for 48 h at 20°C. Adult worms were then dissected in M9 medium (6 g/l $Na_2HPO_4$, 3 g/l $KH_2PO_4$, 5 g/l NaCl, 0.25 g/l $MgSO_4$) and five embryos were transferred onto a 2% agarose pad and recorded simultaneously from the one-cell stage to four cells. Time-lapse images were recorded at 24°C as described previously (Son-neville *et al*, 2017) taking images every 10 s, either using an Olympus IX81 microscope (MAG Biosystems) with a CSU-X1 spinning disc confocal imager (Yokogawa Electric Corporation) and a Cascade II camera (Photometrics), or using a Zeiss Cell Observer SD microscope with a Yokogawa CSU-X1 spinning disc and a HAMA-MATSU C13440 camera, fitted with a PECON incubator. Both microscopes utilised a 60×/1.40 Plan Apochromat oil immersion lens (Olympus). A single optical section (z-layer) was imaged for each time point.

Images were captured using MetaMorph software (Molecular Devices) or using the ZEN blue software (Zeiss) and analysed with ImageJ software (National Institute of Health). For each time-lapse experiment depicted in the figures, the raw images for selected time points were rotated in order to orient the anterior of the chosen embryo to the left and then cropped to focus on a particular nucleus or nuclei, or on the entire embryo. Each series of images was then combined into a contiguous sequence, and the images were subjected to Gaussian Blur with a radius of 1 pixel. Subsequently, the "levels" were adjusted, the pixel density was adjusted to 300 dots per inch and the "bit depth" was changed from 16-bits to 8-bits per channel. Images were processed in a similar manner in order to generate videos, except that time points were not combined into a sequence and the pixel density was not adjusted to 300 dpi.

## Synthetic lethality analysis in *Caenorhabditis elegans*

RNAse III-deficient HT115 bacteria were transformed with an L4440-derived plasmid, corresponding to the required RNAi treatment. For the experiment in Fig 6A and C, the RNAi dose was titrated by mixing the indicated proportion of bacterial cultures expressing *tim-1* and *ubxn-3* double-stranded RNA or containing an empty plasmid. For the E2 screening experiment in Fig 1, the indicated proportion of bacterial cultures expressing each E2 group was mixed as indicated with bacteria expressing *ubxn-3* double-stranded RNA or containing an empty plasmid. For the experiment in Fig EV5, bacterial cultures expressing *lrr-1* or *tim-1* RNAi was used as indicated to feed *trul-1Δ* worms. For the experiment in Appendix Fig S1G and H, bacterial culture containing an empty plasmid was used as indicated to feed *ubc-3Δ, ubc-7Δ, ubc-3Δ ubc-7Δ or ubxn-3Δ* worms. All cultures were grown to OD600 = 1, and worms were then incubated on RNAi feeding plates for 48 h at 20°C. For each condition, triplicate experiments were performed, in each of which 5 adult worms were allowed to produce embryos on a plate during a period of 180 min, after which the adults were removed, and the embryos were counted. Two days later, the number of embryos that had developed into viable adults was determined (between 50 and 120 embryos for each set of embryos from 5 worms). Embryonic viability was expressed as the ratio between the number of viable embryos and the total number of embryos, and the average and standard deviation were then determined for each triplicate set.

## Brood size analysis in *Caenorhabditis elegans*

To determine the number of progeny per adult, worms were grown at 20°C. For each genotype, five L4 larvae were singled on NGM plates with "OP50 bacteria" and the worms were transferred onto a new plate every 24 h for 3 days and then removed. Adult progeny were counted 2–4 days after being laid. The number of viable progeny for each genotype was determined as the sum of the progeny on three plates. The average brood size was then determined for each genotype in three independent experiments, together with the standard deviation from the mean.

## Protease inhibitor cocktails

The following cocktails of protease inhibitors were used as indicated in the sections below:

### Protease inhibitor cocktail 1
One Roche EDTA-free protease inhibitor tablet (000000011873580001, Roche) per 25 ml of buffer (one tablet dissolved in 1 ml water makes a 25× stock solution), plus 1 ml of Sigma protease inhibitor cocktail (P8215, Sigma-Aldrich) per 100 ml of buffer.

### Protease inhibitor cocktail 2
One Roche EDTA-free protease inhibitor tablet (000000011873580001, Roche) per 25 ml of buffer (one tablet dissolved in 1 ml water makes a 25× stock solution), plus 1 ml of Sigma protease inhibitor cocktail (P8215, Sigma-Aldrich) per 100 ml of buffer, together with 0.5 mM PMSF, 5 mM benzamidine HCl, 1 mM AEBSF (A8456, Sigma-Aldrich) and 1 mg/ml Pepstatin A (P5318, Sigma-Aldrich).

### Protease inhibitor cocktail 3

One Roche EDTA-free protease inhibitor tablet (000000011873580001, Roche) per 25 ml of buffer (one tablet dissolved in 1 ml water makes a 25× stock solution), plus 0.5 mM PMSF.

## Extracts of worm embryos and immunoprecipitation of worm replisome

RNAse III-deficient HT115 bacteria were transformed with an L4440-derived plasmid, corresponding to the required RNAi treatment. A 10 ml pre-culture was then grown overnight and used to inoculate a 450 ml culture in "Terrific Broth" (12 g/l Tryptone, 24 g/l yeast extract, 9.4 g/l $K_2HPO_4$, 2.2 g/l $KH_2PO_4$, adjusted to pH 7). After 7 hours of growth in a baffled flask at 37°C with agitation, expression of dsRNA was induced overnight at 20°C by addition of 3 mM IPTG. The bacteria were then pelleted and resuspended with one-fifth volume of buffer (M9 medium supplemented with 75 mg/l cholesterol; 100 mg/l ampicillin; 50 mg/l tetracycline; 12.5 mg/l amphotericin B; 3 mM IPTG).

For each experiment, 1 ml of a synchronised population of L4 worms expressing GFP-PSF-1 or TAP-PSF-1 were fed for 50 h at 20°C on a 15-cm RNAi plate (see above), supplemented with 10 g of bacterial pellet for the required RNAi treatment, prepared as described above. After feeding, the adult worms were washed in M9 medium and resuspended for 2 min at room temperature in 14 ml of "bleaching solution" (for 100 ml: 36.5 ml $H_2O$, 45.5 ml 2 M NaOH and 7 ml ClNaO 10%), then pelleted for 1 min at 300 g. This bleaching procedure was repeated two more times, corresponding to a total of 12 min in bleaching solution, in order to lyse the adult worms and release embryos (about 0.6–0.8 g). After bleaching, the embryos were washed twice with M9 medium.

The remaining steps were performed at 4°C and are modified from our previously described methods (Sonneville *et al*, 2017). Embryos were washed twice with lysis buffer (100 mM HEPES-KOH pH 7.9, 100 mM potassium acetate, 10 mM magnesium acetate, 2 mM EDTA, 0.02% IGEPAL CA-630, 10% glycerol) and then resuspended with three volumes of lysis buffer that was supplemented with 2 mM sodium fluoride, 2 mM sodium β-glycerophosphate pentahydrate, 1 mM dithiothreitol (DTT), 1× Protease Inhibitor Cocktail 1 and 5 μM Propargylated ubiquitin to inhibit de-ubiquitylase enzymes (kindly provided by Axel Knebel and Clare Johnson; DU49003, MRC PPU reagents and services). The mixture was transferred dropwise into liquid nitrogen to prepare "popcorn", which was stored at −80°C. We then ground ~2.5 g of the frozen popcorn in a SPEX SamplePrep 6780 Freezer/Mill. After thawing, we added one-quarter volume of lysis buffer (with additional 1 mM DTT, 2 mM sodium fluoride, 2 mM sodium β-glycerophosphate pentahydrate, plus 1× protease inhibitor cocktail 1). Chromosomal DNA was digested with 1600 U of Universal Nuclease (Pierce™, 88702, Thermo Fisher Scientific) for 30 min at 4°C. Extracts were centrifuged at 25,000 g for 30 min and then for 100000 x g for 1 h.

For immunoprecipitation of GFP-PSF-1 (Figs 1G and EV1), the extract was pre-incubated with agarose beads (0.4 ml slurry; Protein G agarose Fast Flow, PCA-G1000, Generon) for 30 min. Immunoprecipitation of TAP-SLD-5 (Figs 5 and EV5C) did not require such a pre-depletion step. At this point, 50 μl of extract was added to 100 μl of 1.5× Laemmli buffer and stored at −80°C. The remaining ~2 ml of extract was then incubated for 90 min

with 40 μl slurry of GFP-Trap_A beads (gta-20, Chromotek) or 200 μl slurry of magnetic beads (Dynabeads M-270 Epoxy; 14302D, Thermo Fisher Scientific) coupled to rabbit immunoglobulin G (S1265, Sigma-Aldrich) as described below. The beads were washed four times with 1 ml of wash buffer (lysis buffer supplemented with 1 mM DTT, 2 mM sodium fluoride, 2 mM sodium β-glycerophosphate pentahydrate, plus 1× protease inhibitor cocktail 1) and the bound proteins were eluted at 95°C for 5 min in 100 μl of 1× Laemmli buffer and stored at −80°C. Alternatively, proteins associated with TAP-SLD-5 were eluted from the beads by incubation with 80 μl of wash buffer (without protease inhibitors) plus 20 units AcTEV protease (12575015, Thermo Fisher Scientific) at 20 °C for 1 h, before addition of 40 μl 3× Laemmli buffer and heating at 95°C for 5 min.

## Summary of protein purification and associated buffers

Proteins purified in this study are listed in Appendix Table S1. Ubiquitin (and mutated variants), TEV protease (DU6811, MRC PPU Reagents and Services) and PreScission (DU34905, MRC PPU Reagents and Services) protease were kindly provided by Dr. Axel Knebel. The other proteins were produced as described in Appendix Materials and Methods, using the buffers below:

Buffer A: 25 mM Hepes-KOH pH 7.6, 10% glycerol, 0.02% IGEPAL CA-630, 1 mM DTT.
Buffer B: 25 mM Hepes-KOH pH 7.6, 10% glycerol, 0.02% IGEPAL CA-630, 1 mM TCEP.
Buffer C: 25 mM Tris-Cl pH 8.6, 10% glycerol, 0.02% IGEPAL CA-630, 1 mM TCEP.
Buffer D: 25 mM Bis-Tris-Cl pH 6.8, 10% glycerol, 0.02% IGEPAL CA-630, 1 mM TCEP.
Buffer E: 25 mM Hepes-KOH pH 7.6, 10% glycerol, 0.02% Tween 20, 1 mM DTT.
Buffer F: 25 mM Hepes-KOH pH 7.6, 10% glycerol, 1 mM CHAPS, 1 mM TCEP.
Buffer G: 25 mM Hepes-KOH pH 7.6, 40% glycerol, 0.02% IGEPAL CA-630, 1 mM DTT.
Buffer H: 50 mM Tris–HCl pH 8.0, 0.5 mM TCEP.

## Expression of proteins in budding yeast

The *Saccharomyces cerevisiae* strains used in this study are shown in Appendix Tables S1–S2. Yeast cells were grown at 25°C in YP medium (1% Yeast Extract, 21275, Becton Dickinson; 2% bacteriological peptone, LP0037B, Oxoid) supplemented with 2% Raffinose. In each case, a 12-litre exponential culture was grown to 2-3 x $10^7$ cells / ml and then induced for 6 h at 20°C by addition of galactose to a final concentration of 2%. Cells were collected by centrifugation and washed once with lysis buffer (indicated below for each purification) lacking protease inhibitors. Cell pellets (~ 30 g) were then resuspended in 0.3 volumes of the indicated lysis buffer containing protease inhibitors. The resulting suspensions were then frozen dropwise in liquid nitrogen and stored at − 80°C. Subsequently, the entire sample of frozen yeast cells were ground in the presence of liquid nitrogen, using a SPEX CertiPrep 6850 Freezer/Mill with 6 cycles of 2' at a rate of 15. The resulting powders were then stored at −80°C. Details for each protein are given in Appendix Materials and Methods.

**Expression of proteins in bacterial cells**

The plasmids for bacterial expression used in this study were shown in Appendix Tables S1 and S3. Each plasmid was transformed into Rosetta (DE3) pLysS (70956, Novagen), which was grown in LB medium supplemented with 50 µg /ml ampicillin (pET15b based plasmids) or 50 µg/ml kanamycin (pK27SUMO based plasmids). Subsequently, a 10 ml culture was grown overnight at 37°C with shaking at 200 rpm. The following morning, the culture was diluted 50-fold into 500 ml of selective medium and then left to grow at 37°C until an OD600 of 1 was reached. At this point, 1 mM IPTG was added and expression was induced overnight at 18°C. Cells were harvested by centrifugation for 10 min in a JLA-9.1000 rotor (Beckman) at 5,180 $g$. The cell pellets were then stored at −80°C. Details for each protein are given in Appendix Materials and Methods.

**In vitro CMG ubiquitylation assays**

Reactions (typically 10 µl in volume) containing 25 mM Hepes-KOH (pH 7.6), 0.02% IGEPAL CA-630, 0.1 mg/ml BSA, 1 mM DTT, 10 mM Mg(OAc)$_2$, 10 µM ubiquitin, 5 mM ATP and 3.3 µl protein mix were assembled on ice. The protein mix contained 300 mM KOAc, so that the final KOAc concentration of the reaction was 100 mM. The components of the protein mixture are indicated for each experiment in the figures, and unless specified otherwise, the final concentration of each protein was 50 nM UBA-1, 300 nM LET-70, 300 nM UBC-18, 50 nM ARI-1, Neddylation enzymes (50 nM ULA-1_RFL-1, 300 nM UBC-12, 100 nM DCN-1a and 500 nM NED-8), 300 nM UBC-3, 15 nM CMG, 20 nM CTF-4, 30 nM MCM-10, 30 nM POLε, 30 nM CLSP-1, 60 nM TIM-1_TIPN-1, 60 nM CTF-18_RFC and 15 nM CUL-2$^{LRR-1}$/CUL-2_K719R_K749R$^{LRR-1}$/CUL-2$^{VHL-1}$. Ubiquitylation reactions were conducted at 20°C for 20 min. Reactions were stopped by addition of 20 µl 1.5× Laemmli buffer and heating at 95°C for 5 min.

For the experiments in Figs 2C and D, 3B, E, F, EV2A, Fig EV3B, D, E, G, wild-type ubiquitin was replaced as shown with 10 µM of the indicated mutant ubiquitin proteins.

For the experiments in Fig EV1C, 300 nM of the indicated E2s (UBC-3, UBC-7, UBC-1 and UBC-14) were used as shown with 10 µM of the indicated mutant ubiquitin proteins.

For the experiment in Fig EV1E, the reactions only contained the following proteins: 50 nM UBA-1, 300 nM UBC-3, 15 nM of the indicated E3 ligases (CUL-2$^{LRR-1}$, CUL-2-2R$^{LRR-1}$ and CUL-2$^{VHL-1}$) and 10 µM of FLAG-ubiquitin. Polyubiquitin chains were detected by immunoblotting with anti-FLAG antibodies.

**Immunoprecipitation of reconstituted replisomes**

Reactions (typically 20 µl in volume) containing 25 mM Hepes-KOH (pH 7.6), 0.02% IGEPAL CA-630, 0.1 mg/ml BSA, 1 mM DTT, 10 mM Mg(OAc)$_2$, 5 mM ATP and 6.6 µl protein mix were assembled on ice for 30 min. The protein mix contained 300 mM KOAc, so the final KOAc concentration of the reactions was 100 mM. Each sample was then incubated at 4°C with 2 µl magnetic beads (Dynabeads M-270 Epoxy; 14302D, Thermo Fisher Scientific) that had been coupled to anti-SLD-5 antibody as described below. After 1 hour, protein complexes bound to the magnetic beads were

washed twice with 1 ml of buffer containing 25 mM Hepes-KOH (pH 7.6), 0.02% IGEPAL CA-630, 0.1 mg / ml BSA, 1 mM DTT, 10 mM Mg(OAc)$_2$ and 100 mM KOAc. The bound proteins were eluted at 95°C for 5 min in 30 µl of 1× Laemmli buffer.

For the experiment in Fig 2E, a 10 µl volume of the standard *in vitro* CMG ubiquitylation assay was used for the subsequent immunoprecipitation step. The experiments in Figs 3D and EV3A also involved a 10 µl reaction, with the indicated components corresponding to 15 nM CMG, 20 nM CTF-4, 30 nM MCM-10, 30 nM POLε, 30 nM CLSP-1, 60 nM TIM-1_TIPN-1, 60 nM CTF-18_RFC and 15 nM CUL-2$^{LRR-1}$.

**In vitro CMG disassembly assays**

As described above, CMG ubiquitylation reactions (20 µl in total volume) were assembled at 20°C for 20 min, followed by immunoprecipitation of CMG via 2 µl magnetic beads coupled to anti-SLD-5 antibody. The magnetic beads were then washed twice with 1 ml of buffer containing 25 mM Hepes-KOH (pH 7.6), 0.02% IGEPAL CA-630, 0.1 mg/ml BSA, 1 mM DTT, 10 mM Mg(OAc)$_2$ and 100 mM KOAc. Subsequently, a quarter of the beads (0.5 µl) were resuspended in 15 µl of buffer containing 25 mM Hepes-KOH (pH 7.6), 0.02% IGEPAL CA-630, 0.1 mg/ml BSA, 1 mM DTT, 10 mM Mg(OAc)$_2$, 5 mM ATP, 30 mM KOAc and "unfoldase protein mix" (200 nM CDC-48 with 50 nM UFD-1_NPL-4 and 50 nM UBXN-3/UBXN-3-ΔUBX as indicated in Figs 4C and D, and EV4). Disassembly reactions were then conducted at 20°C for 20 min, with shaking at 1,000 rpm. Subsequently, the beads and associated proteins were isolated using a magnetic rack. The supernatant was removed and combined with 7 µl of 3× Laemmli buffer before heating at 95°C for 5 min. Meanwhile, the beads were washed twice with 0.5 ml of buffer containing 25 mM Hepes-KOH (pH 7.6), 0.02% IGEPAL CA-630, 0.1 mg/ml BSA, 1 mM DTT, 10 mM Mg(OAc)$_2$ and 100 mM KOAc. Finally, the bead-bound proteins were eluted at 95°C for 5 min in 20 µl of 1× Laemmli buffer.

**Glycerol gradient analysis**

Reactions in Fig EV2F (typically 5 µl in volume) containing 25 mM Hepes-KOH (pH 7.6), 200 mM KOAc, 0.02% IGEPAL CA-630, 1 mM DTT, 10 mM Mg(OAc)$_2$, 5 mM ATP, 500 nM DNA substrate (comprising 46 bp double-strand DNA and a 39nt "3'-flap" of single-strand DNA – the sequences are shown in Appendix Table S1) and "protein mix" (100 nM CMG, 50 nM TIM-1_TIPN-1 and 50 nM CUL-2$^{LRR-1}$ were used as indicated) were assembled on ice for 15 min. To assemble glycerol gradients, five different concentrations of glycerol buffers were used (10%, 15%, 20%, 25%, 30%), each containing 25 mM Hepes-KOH (pH 7.6), 200 mM KOAc, 0.02% IGEPAL CA-630, 1 mM DTT, 10 mM Mg(OAc)$_2$, 5 mM ATP. Gradients were assembled by consecutively layering 40 µl of each of the five concentrations of glycerol buffers (30% to 10%) in an ultra-centrifuge tube (P200915MGSG, Beckman). Subsequently, 5 µl of reaction was added to the top of gradient, before spinning for one hour at 249,000 $g$ in a Beckman TLS55 rotor at 4°C. Ten fractions of 20 µl each were then collected from top to the bottom of the gradient. After addition of 10 µl 3× Laemmli buffer, the samples were analysed by SDS–PAGE and immunoblotting.

## Immunoblotting

Protein samples were resolved by SDS–PAGE using the following systems: NuPAGE Novex 4 - 12% Bis-Tris gels (NP0321 and WG1402A, Thermo Fisher Scientific) with NuPAGE MOPS SDS buffer (NP0001, Thermo Fisher Scientific) or NuPAGE MES SDS buffer (NP0002, Thermo Fisher Scientific); NuPAGE Novex 3 - 8% Tris-Acetate gels (EA0375BOX and WG1602BOX, Thermo Fisher Scientific) with NuPAGE Tris-Acetate SDS buffer (LA0041, Thermo Fisher Scientific). The resolved proteins were either stained with colloidal Coomassie blue dye ("Instant Blue", ab119211, Abcam), or else transferred onto a nitrocellulose iBlot membrane (NRO11020-01, Thermo Fisher Scientific) with the iBlot Dry Transfer System (IB1001, Invitrogen), according to the manufacturer's instructions.

The antibodies used for immunoblotting in this study are described in Appendix Table S1. Chemiluminescent signals were detected on Hyperfilm ECL (Amersham, 66601, GE Healthcare) using ECL Western Blotting Detection Reagent (17039552, GE Healthcare). New antibodies that were generated in this study are validated in Appendix Fig S4.

### Preparation of antibody-coated magnetic beads

A slurry of activated magnetic beads (Dynabeads M-270 Epoxy; 14302D, Thermo Fisher Scientific) was prepared by resuspending 300 mg beads in 10 ml dimethyl formamide. Each coupling reaction involved 425 μl slurry of activated magnetic beads, which corresponded to ~ $1.4 \times 10^9$ beads. After removing the supernatant, the beads were washed twice with 1 ml of 0.1 M $NaPO_3$ pH 7.4. Subsequently, the beads were incubated with 300 μg of rabbit immunoglobulin G (S1265, Sigma-Aldrich) or *C. elegans* SLD-5 antibody (SA419, MRC PPU Reagents and Services), 300 μl of 3 M $(NH4)_2SO_4$, plus 0.1 M $NaPO_3$ pH 7.4 up to a total volume of 900 μl. The mixture was then incubated at 4 °C for 2 days with rotation.

Subsequently, the supernatant was removed and the beads were washed four times with 1 ml PBS. The beads were then incubated for 10 min in 1 ml PBS/0.5% IGEPAL CA-630 with rotation at room temperature, before washing twice with 1 ml PBS. Finally, the washed beads were resuspended with 900 μl PBS containing 5 mg/ml BSA.

## Data availability

This study includes no data deposited in external repositories.

**Expanded View** for this article is available online.

## Acknowledgements

We thank Axel Knebel and Clare Johnson for purified UBA-1 and ubiquitin derivatives, Mel Wightman for vectors expressing worm E2 enzymes (UBC-1, UBC-3, UBC-7, UBC-14 and LET-70), Ye Hong for sharing details of the deletions of *C. elegans* tim-1 and clsp-1 genes, Sarah-Lena Offenburger for *C. elegans* cDNA, Anton Gartner for sharing equipment and reagents for *C. elegans* growth and manipulation, and MRC PPU Reagents and Services (https://mrcppureagents.dundee.ac.uk) for antibody production. We are very grateful for financial support from the Medical Research Council (core grant MC_UU_12016/13 to KPML), Cancer Research UK (Programme Grant C578/A24558 to KPML), the Japan Society for the Promotion of Science (reference 201780015 for an award to RF from the JSPS Overseas Challenge Program for Young Researchers) and the Wellcome Trust (reference 204678/Z/16/Z for a Sir Henry Wellcome Postdoctoral Fellowship to TDD).

## Author contributions

All experiments were performed by YX except as follows: the E2 RNAi screen was performed by RF and the key results were then repeated by YX (Fig 1D–F and Fig 1I); RF first characterised the ability of *C. elegans* CDC-48, UFD-1_NPL-4 and UBXN-3 to mediate CMG disassembly; RF also purified Ulp1 SUMO protease, CDC-48.1, UFD-1_NPL-4.1 and UBXN-3 and performed the experiments in Figs 1G and 4A, Appendix Fig S2 and Fig EV3; RNAi experiments involving spinning disc confocal microscopy were performed by RS (Figs 1H and 6B, Appendix Fig S3 and Fig EV5), together with PCR characterisation of *ubc-3Δ*, *ubc-7Δ* and *ubxn-3Δ* (Appendix Fig S1) and immunoblot characterisation of antibodies to CUL-2 and LRR-1 (Appendix Fig S4). RS taught YX and RF all aspects of work with the *C. elegans* system. TDD taught YX to express and purify proteins from budding yeast and then provided expert advice throughout the course of the project. YX, RS and KPML designed the project. KPML and YX wrote the manuscript with critical input from the other authors.

## Conflict of interest

The authors declare that they have no conflict of interest.

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
