## [Review Process File · The EMBO Journal]

TIMELESS-TIPIN and UBXN-3 promote replisome disassembly during replication termination in *C. elegans*

Karim Labib, Yisui Xia, Ryo Fujisawa, Tom Deegan, and Remi Sonnevile
DOI: [10.15252/embj.2021108053](https://doi.org/10.15252/embj.2021108053)

Corresponding author(s): Karim Labib (kpmlabib@dundee.ac.uk) , Remi Sonnevile (rsonnevile@dundee.ac.uk)

Review Timeline:

Submission Date:	17th Feb 21
Editorial Decision:	22nd Mar 21
Revision Received:	18th May 21
Editorial Decision:	9th Jun 21
Revision Received:	10th Jun 21
Accepted:	11th Jun 21

Editor: Hartmut Vodermaier

Transaction Report:

Thank you for submitting your manuscript on *C. elegans* replication termination for our editorial consideration. It has now been reviewed by three expert referees, whose comments are copied below. As you will see, the referees acknowledge the potential interest of your further analyses of replisome disassembly factors in a reconstituted metazoan system, but also raise a number of issues that would need to be clarified prior to publication. Of particular importance would be a better characterization of the in vitro replication system employed (ref 3), addressing the major points raised by referee 2, and improving the overall presentation as agreed on by all referees in their reports and/or the post-review consultations.

Referee #1:

The Labib lab has been investigating the disassembly of CMG helicase complex for many years. In this manuscript, they further elucidated a couple of molecular details of the CMG complex disassembly, one of the essential processes for DNA replication and genome stability. They have found that the TIMELESS-TIPIN complex bridges Cul2-LRR1 to the replisome and this strongly stimulates Cul2-LRR1-dependent ubiquitination of the CMG helicase. In addition, they have also

identified the new role of CDC-48 cofactor UBXN3/FAF1 in the coordination of CDC-48-Ufd1-Npl4 complex for the CMG disassembly. In general, this is a very elegant work, which includes a combination of both in vitro and in vivo approaches to address their scientific questions. Thus, I am very supportive of this manuscript as it addresses important biological questions in the field of DNA replication and p97 system.

However, before this manuscript can be accepted for publication, I would suggest that the authors address several additional questions/comments which will further strengthen the quality of this manuscript.

The most importantly, could the authors directly show that UBXN3's role in the CMG disassembly is linked to the p97/Cdc48 system and ubiquitin. Analysis of UBXN3 variants bearing mutation/deletion in p97 and UB binding domains in one of their phenotypes would directly address my question.

Other comments:

- 1) Why there is no PSF-1 retention on chromatin when *ubc3* and *ubc7* are depleted (Fig 1H), if these are two E2 -conjugating enzymes for Cul2-dependent Ub-chain extension?
- 2) Fig 3B; This experiment should be quantified to claim that the TIM-1-TIPN-1 complex stimulates the priming of Ub-chains on CMG-MCM-7 for 10 fold.
- 3) Molecular weight markers on the gels in Fig 4C and D should be indicated.
- 4) To claim that UBXN3 is important for the disassembly of CMG complex by CDC48-U-N, the authors should demonstrate that UBXN3 variant defective for binding to CDC-48 abolishes the CMG disassembly.
- 5) Could the authors restore siUBXN3/siTIM-1 lethality phenotype by co-depleting MCM7? This will further strengthen their model in vivo (Fig 6C).

Referee #2:

During the termination of DNA replication the replisome is disassembled and this process is important for cell viability. However, many important questions about the mechanism of disassembly remain. In the current manuscript, Xia and colleagues identify the full set of proteins required for replisome disassembly in metazoa and reconstitute this process. The authors also identify a unique role for the replisome complex TIM-TIPIN in promoting replisome disassembly by recruiting the ubiquitin ligase CUL2(Lrr1). Although specific components of the yeast replisome have been shown to stimulate replisome disassembly (Deegan et al, 2020) they are not counterparts of TIM-TIPIN so this result is still interesting and unexpected. However, as impressive as this body of work is overall, much of it appears to be confirmatory based on the authors' previous publications (Sonneville et al 2017, Deegan et al 2020). Additionally, the authors need to provide stronger evidence that the in vivo role of TIM-1/TIPN-1 is direct and the manuscript needs to be re-written to substantially improve clarity.

Major concerns

- A major issue with the manuscript is that it is incredibly difficult to follow. This is due to typos within the manuscript combined with some unclear wording and compounded by experimental complexity (the manuscript describes a new reconstituted system). The manuscript needs to be clear enough for others to follow and for reviewers to fully assess the new reconstituted system. In

the 'minor concerns' listed below I have included a non-exhaustive list of points where additional clarity is needed.

- Some of the results regarding the extent of substrate ubiquitylation could, in principle, be due to contamination by de-ubiquitylating enzymes (DUBs). The authors should show that no DUB activity is present in any component of their reconstitution reactions.
- In Figure 4C, a substantial population of replisomes containing polyubiquitinated MCM-7 are not unloaded (lanes 3 vs 4) suggesting that ubiquitin chain length might not be the only determinant of whether the replisome is disassembled. Thus, while the data presented in Figure 4 are consistent with the 'ubiquitin threshold' previously described in Deegan et al (2020) they are not fully supportive. The authors should adjust their conclusions accordingly.
- Figure 5 shows that knockdown of TIM-1 or TIPN-1 also reduces levels of CLSP-1 and CTF-4 present within the replisome. The authors should address whether the effect of TIM-1/TIPN-1 knockdown could be indirect, due to their involvement in regulating CLSP-1 and CTF-4.
- The role of TIM-1/TIPN-1 in promoting replisome disassembly (Figure 6) could be due to incomplete replication because TIM-1/TIPN-1 are important replication factors. On lines 402-409 the authors note that this is unlikely to be the case because chromosome condensation is not impaired, which the authors use as a read-out for complete replication based on previous work (Sonneville et al, 2015). As it stands, this explanation is unsatisfactory because it is unclear exactly how much replication is required to stimulate chromosome condensation. To use condensation as a read-out for complete replication the authors would need to make a clear case that condensation requires most DNA synthesis to be completed. Alternatively, the authors need to show that TIM-1/TIPN-1 knock down does not interfere with removal of replication proteins that are not targeted by CUL-2_LRR-1 (e.g. RPA).
- The authors state that the mammalian ortholog of UBXN-3, FAF1, is a tumor suppressor (lines 24-26, 528-530). However, the review cited in support of this statement (Menges et al, 2009) only highlights correlative data that raise the possibility FAF1 could be a tumor suppressor. These observations are not strong enough to make a specific claim that the authors' work could lead to the development of future cancer therapies. The authors should remove these claims or cite more recent studies that make a stronger case.

Minor concerns

- There is no clear statement in the introduction that disassembly of the replisome leads to its dissociation from DNA
- The manuscript switches between naming conventions (e.g. CDC-48_UFD-1_NPL4 vs TIMELESS-TIPIN), which is confusing
- The manuscript refers to Star methods and a Key Resources table. Neither of these appear to be part of the manuscript
- Line 400 refers to figure 3A-B. Shouldn't it be Figure 5A-B?
- 'Replisome' refers to all components except for CMG in Fig 2E, while in Fig 2B-D 'replisome' includes CMG

- Could the authors comment on the stoichiometry of the multi-subunit complexes they have purified (Figure 2A)?
- Figure 2B is initially discussed (lines 205-224) without commenting on the enhanced ubiquitylation observed with LET-70 vs ARI-1 plus UBC-18. This difference is quite striking and should really be discussed as the data is introduced.
- Does the K48-linked ubiquitylation on Lines 229-231 refer only to ubiquitylation initiated by UBC-18 and ARI1? The LET-70 ubiquitylation appears to be fairly independent of K48 (lanes 6 vs 9, Fig 2D).
- Lines 231-233 should more clearly explain how Figure 2B-C demonstrate that the major reaction product is a single polyubiquitin chain
- Where is the UBC-1 and UBC-14 data referenced on lines 240-242?
- Neddylation of Cul2 as described on lines 243-246 should be clearly explained somewhere in the manuscript
- Lines 284-288 should make the connection between TIMELESS-TIPIN/CTF4/CLASPIN and the yeast proteins in Baretic et al (2020) clearer.
- In Figure 3F it appears that PSF-3 is ubiquitinated. A band present immediately above the non-specific band in conditions 2 and 3 but not condition 1. This should be commented on.

Referee #3:

Xia et al. present significant progress on how metazoa disassemble replisomes during replication termination. Important players of the pathway are identified and characterised by loss-of-function approaches in *C. elegans* and their mechanistic roles investigated using extensive biochemical reconstitution. The authors also present a first step in re-constituting parts of the metazoan replisome from purified components. In addition, specific ubiquitylation by Cul2-LRR1 of Mcm7 and CMG extraction by p97 are re-constituted. These *in vitro* reactions could be key to advance our understanding in future studies of the molecular machines involved. The work also provides advance in our understanding of how the p97 segregase works (the specific advance presented on ubiquitylation and segregation is harder for me to judge due to my area of expertise).

The present study is timely. It is consistent with earlier work of the Labib lab and other labs in yeast, *Xenopus* and *C. elegans* showing how CMG is disassembled during termination and in the subsequent mitosis. The present study extends and confirms important concepts suggested before mainly in yeast, but also presents unexpected aspects on mechanism and *in vivo* function.

The study presents an enormous amount of high-quality work. The data presented seems flawless. The conclusions are clear and sufficiently supported by the experimental evidence presented.

I strongly support acceptance of the manuscript upon addressing a few points.

Issues to be addressed:

- The biochemistry appears to suggest a linear relationship of the action of Timeless-Tipin and UBXN3: CMG extraction depends on Timeless-Tipin-mediated ubiquitylation, which is a prerequisite for UBXN3-dependent CMG extraction by p97. The functional experiments in Fig 6 suggest synergistic effects of simultaneously inactivating both pathways, which rather fits a model where the action of Timeless-Tipin and UBXN3 occurs in parallel pathways. Please present an explanation for this potential contradiction.

- Re-constituted partial C elegans replisomes are used to re-constitute replisome-dependent Mcm7 ubiquitylation and disassembly. A characterisation of the partial replisomes is required to be able to judge the approach used. Please provide such characterisation, for example by showing association of the replisome components with CMG. To my view, some (perhaps very basic) evidence will be important even if the authors plan to publish this C elegans replisome system separately. One reason why this is important is that these replisomes were not formed by origin-dependent replication initiation, but in an "artificial" way by origin-independent assembly, as it has been done successfully for yeast replisomes by the O'Donnell lab, for example.

- Fig 3: From line 290 the authors state 'In contrast, removal of CTF-4 or CLSP-1 had no impact on the efficiency of CMG ubiquitylation' A control is required to show that CTF-4 and CLSP-1 associate with the replisome.

- From line 310 the authors write "These data likely reflect a direct interaction of Cul-2-LRR1 with TIM-1-TIPN-1 in the context of the replisome". Please test and state whether an interaction of Cul-2-LRR1 and TIM-1-TIPN-1 occurs outside the context of full replisomes, which would be more direct evidence for a direct interaction.

- Line 463: "...likely reflecting the associated defect in CMG helicase disassembly during DNA replication termination".

I am not sure there is sufficient evidence for this conclusion. Can an alternative model be excluded where synthetic lethality results from defective replication termination-dependent disassembly in conjunction with a simultaneous defect in disassembly in the subsequent mitosis? Please elaborate or present this alternative explanation for the synthetic lethality.

minor issues:

- Introduction line 74 to line 95. This paragraph is hard to understand and should be revised.

- line 112: typo: identify should be identity

- line 139: Is C elegans ubc-3 the only orthologue of human UBE2R1 and 2 (line 133)? Please make a statement.

- line 168: "...and with other E2 enzymes." How do you conclude this? Because the effect of Ubc3 and Ubc7 RNAi is not as strong as with LRR1 siRNA? Please specify.

- line 270/71: Specify the experiment made by changing to "However, the association of...that lack all other replisome components except CMG"

- - line 360: typo: UBNX-3

We thank all three reviewers for their comments and their interest in our work. In order to address all of the requested changes, the revised version of our manuscript includes 12 new experiments (Figure 4A, Figure EV1C, Figure EV2A, Figure EV2B, Figure EV2C, Figure EV2F, Figure EV2H, Figure EV3, Figure EV4C, Figure EV5, Appendix Figure S1G-H, Appendix Figure S3), together with extensive changes to the text. These are discussed in detail below.

Referee #1:

“The Labib lab has been investigating the disassembly of CMG helicase complex for many years. In this manuscript, they further elucidated a couple of molecular details of the CMG complex disassembly, one of the essential processes for DNA replication and genome stability. They have found that the TIMELESS-TIPIN complex bridges Cul2-LRR1 to the replisome and this strongly stimulates Cul2-LRR1-dependent ubiquitination of the CMG helicase. In addition, they have also identified the new role of CDC-48 cofactor UBXN3/FAF1 in the coordination of CDC-48-Ufd1-Npl4 complex for the CMG disassembly. In general, this is a very elegant work, which includes a combination of both in vitro and in vivo approaches to address their scientific questions. Thus, I am very supportive of this manuscript as it addresses important biological questions in the field of DNA replication and p97 system. However, before this manuscript can be accepted for publication, I would suggest that the authors address several additional questions/comments which will further strengthen the quality of this manuscript. The most importantly, could the authors directly show that UBXN3's role in the CMG disassembly is linked to the p97/Cdc48 system and ubiquitin. Analysis of UBXN3 variants bearing mutation/deletion in p97 and UB binding domains in one of their phenotypes would directly address my question.”

The data in Figure 4C showed directly that UBXN-3 stimulates CMG disassembly by CDC-48 (new data in Figure EV3 reproduce this effect), dependent on its well characterised ubiquitin receptors UFD-1 and NPL-4. To confirm that this role of UBXN-3 is indeed linked to CDC-48, we repeated the above experiment and included a mutant of UBXN-3 that lacked the UB domain that is required for association with CDC-48. The new data are shown in the new version of Figure EV3 and illustrate that UBXN-3- Δ UBX does not support CMG disassembly. We also now show that the reaction stimulated by UBXN-3 is dependent upon poly-ubiquitylation via lysine 48 of ubiquitin, since CMG disassembly is blocked in reactions utilising K48R ubiquitin or K0 ubiquitin (lysine free), despite the presence of all disassembly factors (Figure EV3).

The reviewer suggested using UBXN-3 variants *in vivo* to analyse CMG disassembly phenotypes, but this was not possible for several reasons. We had already shown in Appendix Figure S1 that *ubxn-3 Δ* is lethal (Figure S1G-H). Therefore, loss of function mutations in *ubxn-3* should also cause lethality, precluding any detailed physiological analysis. An alternative approach might have been to make RNAi-resistant versions of *ubxn-3* mutants, but this is not practical in *C. elegans*, where RNAi is based on targeting large regions of the gene, making it extremely difficult to design RNAi-resistant forms.

Other comments:

1) **“Why there is no PSF-1 retention on chromatin when *ubc3* and *ubc7* are depleted (Fig 1H), if these are two E2 -conjugating enzymes for Cul2-dependent Ub-chain extension?”**

Although previous work with human cells, cited in our original manuscript (lines 135-138), indicates that the human orthologues of worm UBC-3 (UBE2R1/R2) and UBC-7 (UBE2G1) are sufficient to explain chain elongation for substrates of cullin ligases, it appears that the situation in *C. elegans* is a bit more complicated. We now include new data in Appendix Figure S1 to show that *ubc-3Δ ubc-7Δ* worms are viable (Appendix Figure S1G), though with reduced brood size (Appendix Figure S1H) and developmental abnormalities (our unpublished data). This indicates that other E2 enzymes must be able to compensate partially in the absence of UBC-3 and UBC-7, thereby explaining why PSF-1 retention is not seen upon double RNAi to *ubc-3* and *ubc-7*.

Nevertheless, our data show that double depletion of UBX-3 and UBC-7 leads to a synthetic lethal phenotype in combination with *ubxn-3* RNAi (Figure 1F), associated with accumulation of partially ubiquitylated CMG (Figure 1G). Moreover, both UBC-3 and UBC-7 support the elongation of K48-linked ubiquitin chains on CMG-MCM-7 *in vitro*, in combination with CUL-2^{LRR-1} (new data for UBC-7 are presented in Figure EV1C). These findings indicate that UBC-3 and UBC-7 are indeed important players in the ubiquitylation of cullin ligase substrates in *C. elegans*.

2) **“Fig 3B; This experiment should be quantified to claim that the TIM-1-TIPN-1 complex stimulates the priming of Ub-chains on CMG-MCM-7 for 10 fold.”**

We thank the reviewer for pointing this out. The original version of the text was based on the fact that CMG ubiquitylation in Figure 3B lane 2 (presence of TIM-1_TIPN-1 and 1 nM CUL-2^{LRR-1}) appeared similar to CMG ubiquitylation in Figure 3B lane 9 (absence of TIM-1_TIPN-1 and 9 nM CUL-2^{LRR-1}).

We now include three repeats of this experiment in Figure EV2B, together with a quantification of these new data in Figure EV2C. The data indicate that TIM-1_TIPN-1 stimulate CMG ubiquitylation several fold, but not quite ~10-fold as previously estimated. Therefore, we have adjusted the text accordingly, to say **“the priming of ubiquitin chains on CMG-MCM-7 was stimulated several-fold in the presence of TIM-1_TIPN-1”** (lines 320-321).

3) **“Molecular weight markers on the gels in Fig 4C and D should be indicated.”**

We have now done this.

4) **“To claim that UBXN3 is important for the disassembly of CMG complex by CDC48-U-N, the authors should demonstrate that UBXN3 variant defective for binding to CDC-48 abolishes the CMG disassembly.”**

As discussed above (Major Point), we now include new data to illustrate this point, in Figure EV3.

5) **“Could the authors restore *siUBXN3/siTIM-1* lethality phenotype by co-depleting MCM7? This will further strengthen their model *in vivo* (Fig 6C).”**

This is an interesting idea but in practice is a very challenging experiment. Depleting MCM-7 is highly toxic - even if the bacteria expressing *mcm-7* RNAi are ‘diluted’ 50-fold, by mixing with bacteria carrying empty vector, then this still causes 100% lethality in worms. In other words, even 2% *mcm-7* RNAi is fully lethal in *C.*

elegans. Therefore, it is very difficult to try and exploit the reduction of CMG complexes upon depletion of MCM-7, to see whether this might suppress lethality of *ubxn-3 tim-1* double RNAi, when depletion of MCM-7 itself compromises genome duplication very significantly.

For this reason, there wasn't really a predicted result for this experiment, as it wasn't clear whether the potentially 'good' effects of reducing MCM-7 in this particular context, would or should predominate over the clearly very bad effects of depleting MCM-7 in general. We did try the suggested experiment several times, but we didn't observe any suppression of the lethality of *ubxn-3 tim-1* RNAi, in combination with 0.5 % or 1% RNAi to *mcm-7*. It is not really possible to interpret this result.

Referee #2:

“During the termination of DNA replication the replisome is disassembled and this process is important for cell viability. However, many important questions about the mechanism of disassembly remain. In the current manuscript, Xia and colleagues identify the full set of proteins required for replisome disassembly in metazoa and reconstitute this process. The authors also identify a unique role for the replisome complex TIM-TIPIN in promoting replisome disassembly by recruiting the ubiquitin ligase CUL2(Lrr1). Although specific components of the yeast replisome have been shown to stimulate replisome disassembly (Deegan et al, 2020) they are not counterparts of TIM-TIPIN so this result is still interesting and unexpected. However, as impressive as this body of work is overall, much of it appears to be confirmatory based on the authors' previous publications (Sonneville et al 2017, Deegan et al 2020). Additionally, the authors need to provide stronger evidence that the *in vivo* role of TIM-1/TIPN-1 is direct and the manuscript needs to be re-written to substantially improve clarity.”

As discussed below, we now provide a range of additional evidence that indicates that the *in vivo* role of TIM-1_TIPN-1 in CMG ubiquitylation is indeed direct, consistent with our *in vitro* experiments with the reconstituted CMG ubiquitylation system. We have also made an extensive series of alternations to the manuscript to improve clarity.

Major concerns

- “A major issue with the manuscript is that it is incredibly difficult to follow. This is due to typos within the manuscript combined with some unclear wording and compounded by experimental complexity (the manuscript describes a new reconstituted system). The manuscript needs to be clear enough for others to follow and for reviewers to fully assess the new reconstituted system. In the 'minor concerns' listed below I have included a non-exhaustive list of points where additional clarity is needed.”

We apologise for typos and lack of clarity over wording and have now done our best to address these issues, including changes in response to all of the points listed below in minor comments (where we give details of the alterations that we have made). We have also made extensive additions to the Appendix Table S1 (reagents and resources from this study) and to Materials and Methods, in order to improve clarity.

- “Some of the results regarding the extent of substrate ubiquitylation could, in principle, be due to contamination by de-ubiquitylating enzymes (DUBs). The authors should show that no DUB activity is present in any component of their reconstitution reactions.”

We note that the main conclusions from the *in vitro* work are all substantiated by the *in vivo* work. The reviewer did not specify which “**results regarding the extent of substrate ubiquitylation**” she / he was concerned about, but to illustrate that our *in vitro* data are not due to DUB contamination, we did the following experiment (see panel A of Figure for Reviewer 2 (NOT FOR PUBLICATION)):
- we made a mixture of all the proteins in this study that were expressed in budding yeast

- we made a mixture of all the proteins in this study that were expressed in bacteria (we note that bacteria lack DUBs)
- we performed a CMG ubiquitylation reaction as in Figure 2B lane 8 of our manuscript, then isolated ubiquitylated CMG away from the reaction components by IP of SLD-5, followed by extensive washing (firstly with high salt to help remove reaction components, then with low salt at the end).
- we then incubated the bead-bound ubiquitylated CMG with:
(i) human USP2 (a well-characterised DUB) as a positive control.
(ii) the mixture of yeast expressed proteins
(iii) the mixture of bacterially expressed proteins.
All reactions were performed in duplicate, with one half (lanes 5-8) also containing the DUB inhibitor Ub-Prg, in order to show the specificity of any observed deubiquitylation.

As shown in panel B of 'Figure for Reviewer 2 (NOT FOR PUBLICATION)', the USP2 control removed poly-ubiquitin chains from CMG-MCM-7 (lane 4) and this activity was blocked by the DUB inhibitor (lane 8). In contrast, the proteins used in our study did not show DUB activity.

- "In Figure 4C, a substantial population of replisomes containing polyubiquitinated MCM-7 are not unloaded (lanes 3 vs 4) suggesting that ubiquitin chain length might not be the only determinant of whether the replisome is disassembled. Thus, while the data presented in Figure 4 are consistent with the 'ubiquitin threshold' previously described in Deegan et al (2020) they are not fully supportive. The authors should adjust their conclusions accordingly."

We have repeated the experiment in Figure 4C and have included the new data in the revised version (as Figure EV3). The new data confirm that MCM-7 with 1 to ~5 ubiquitins is retained preferentially on the beads, whereas MCM-7 with progressively longer chains (>5 ubiquitins) is preferentially released into the supernatant. The same applies if we use *C. elegans* CDC-48_UFD-1_NPL-4_UBXN-3 to disassemble ubiquitylated yeast CMG helicase (our unpublished data). Therefore, the 'ubiquitin threshold' is clearly a conserved feature of CMG disassembly by CDC-48 / p97 and the same is likely to apply to the disassembly of other ubiquitylated protein complexes by CDC-48 / p97.

It is important to note that we do not claim that the ubiquitin threshold is the only determinant of disassembly by CDC-48 / p97. Indeed, a key aspect of our manuscript is the identification of a new determinant of the disassembly of ubiquitylated CMG by *C. elegans* CDC-48_UFD-1_NPL-4, namely the stimulation of CDC-48_UFD-1_NPL-4 by UBXN-3. We cannot exclude still further determinants of the function of metazoan CDC-48_UFD-1_NPL-4, but it is also important to note that there was no reason to expect that the reconstituted CMG disassembly reaction should go to completion under these particular *in vitro* conditions. From the point of view of our manuscript, the important point was not the overall efficiency of disassembly (dependent on many variables). Instead, the two key points were that there is a clear preference for disassembling CMG complexes with more than ~5 ubiquitins conjugated to MCM-7, and a clear role for UBXN-3 in stimulating CDC-48_UFD-1_NPL-4.

- "Figure 5 shows that knockdown of TIM-1 or TIPN-1 also reduces levels of CLSP-1 and CTF-4 present within the replisome. The authors should address

whether the effect of TIM-1/TIPN-1 knockdown could be indirect, due to their involvement in regulating CLSP-1 and CTF-4.”

Figure EV4 shows that CRISPR-Cas9 deletion of the *ctf-4* gene locus, removing 89% of the *ctf-4* coding sequence (Figure EV4A-B), has no impact on the ubiquitylation of CMG-MCM-7 *in vivo* (Figure EV4C, compare lanes 5 and 7). Furthermore, we now also show that RNAi depletion of *clsp-1* in *ctf-4* Δ worms does not inhibit the ubiquitylation of CMG-MCM-7, or the association of CUL-2^{LRR-1} with CMG (Figure EV4C, lane 8).

Therefore, the effect of *tim-1* RNAi *in vivo* (reduction of CMG-MCM-7 ubiquitylation and the displacement of CUL-2^{LRR-1} from the replisome) cannot be explained via indirect effects of TIM-1 depletion on CTF-4 or CLSP-1. Most importantly, our *in vitro* data show directly that TIM-1_TIPN-1, but not CTF-4 or CLSP-1, is important for CMG-MCM-7 ubiquitylation (Figure 3A). Moreover, TIM-1_TIPN-1 stimulates the association of CUL-2^{LRR-1} with CMG *in vitro*, whereas CTF-4 and CLSP-1 are dispensable for the interaction of E3 ligase with helicase (Figure 3D). These data most likely reflect a direct interaction of CUL-2^{LRR-1} with both CMG and TIM-1_TIPN-1. Correspondingly, we now show that CUL-2^{LRR-1} co-migrates in a glycerol gradient with CMG and TIM-1_TIPN-1 (new data in Figure EV2F). The interaction of CUL-2^{LRR-1} with either CMG or TIM-1_TIPN-1 is dependent upon the presence of all three factors, likely reflecting a ternary complex that stabilises the individual interactions between the ligase, helicase and TIM-1_TIPN-1. Consistent with these data, TIM-1_TIPN-1 and MCM-7 are the only detectable substrates of CUL-2^{LRR-1} in the worm replisome (Figure 3E-F and Figure EV2G), further indicating a direct interaction of CUL-2^{LRR-1} with both CMG and TIM-1_TIPN-1. In addition, we now show that CUL-2^{LRR-1} can ubiquitylate TIM-1_TIPN-1 even in the absence of CMG or other replisome factors (Figure EV2H, lane 3). Nevertheless, ubiquitylation of TIM-1_TIPN-1 is more efficient in the presence of CMG (Figure EV3H, lane 1), supporting the idea that ternary complex formation stabilises the pair-wise interactions of CUL-2^{LRR-1} with both CMG and TIM-1_TIPN-1.

- “The role of TIM-1/TIPN-1 in promoting replisome disassembly (Figure 6) could be due to incomplete replication because TIM-1/TIPN-1 are important replication factors. On lines 402-409 the authors note that this is unlikely to be the case because chromosome condensation is not impaired, which the authors use as a read-out for complete replication based on previous work (Sonneville et al, 2015). As it stands, this explanation is unsatisfactory because it is unclear exactly how much replication is required to stimulate chromosome condensation. To use condensation as a read-out for complete replication the authors would need to make a clear case that condensation requires most DNA synthesis to be completed. Alternatively, the authors need to show that TIM-1/TIPN-1 knock down does not interfere with removal of replication proteins that are not targeted by CUL-2_LRR-1 (e.g. RPA).”

The first version of the manuscript already showed that RNAi inactivation of *tim-1* does not cause a severe drop in embryonic viability (Figure 6A), indicating that *tim-1* RNAi does not lead to incomplete DNA synthesis. Similarly, the data in Figure 5A shows that equivalent amounts of CMG are present in control worms or worms treated with *tim-1* RNAi (compare levels of MCM-2 and CDC-45 in IPs of TAP-PSF-1, lanes 5 and 8), whereas we previously showed that incomplete DNA replication leads to lower levels of CMG (e.g Supplementary Figure 1f of Sonneville et al 2017, Nat. Cell Biol., compare levels of MCM-2 and CDC-45 in 3rd and 4th lanes).

We now also present new data in Figure EV5 to show that *tim-1* RNA does not lead to accumulation of the single-strand DNA binding protein RPA on chromatin, in contrast to the effect of depleting replication factors such as DNA polymerase alpha. As discussed in the revised text (lines 449-463), these data argue that *tim-1* RNAi does not arrest DNA synthesis but instead impairs CMG ubiquitylation during DNA replication termination. These data are consistent with our *in vitro* data that show directly that TIM-1_TIPN-1 is important for the efficiency of CMG-MCM-7 ubiquitylation and the association of CUL-2^{LRR-1} with the CMG helicase.

- ***“The authors state that the mammalian ortholog of UBXN-3, FAF1, is a tumor suppressor (lines 24-26, 528-530). However, the review cited in support of this statement (Menges et al, 2009) only highlights correlative data that raise the possibility FAF1 could be a tumor suppressor. These observations are not strong enough to make a specific claim that the authors' work could lead to the development of future cancer therapies. The authors should remove these claims or cite more recent studies that make a stronger case.”***

We take the reviewer's point and have now cited (on line 585) a recent study (Bonjoch et al, 2020) that indicates that germline mutations in the human *FAF1* gene are associated with a hereditary form of colorectal cancer.

Minor concerns

- ***“There is no clear statement in the introduction that disassembly of the replisome leads to its dissociation from DNA.”***

Lines 53-56 of the revised manuscript now state: ***“Cdc48 / p97 then unfolds ubiquitylated MCM7 (Deegan et al., 2020), leading to the irreversible dissociation of CMG into its component parts and thus to replisome disassembly and the dissociation of replisome components from DNA.”***

- ***“The manuscript switches between naming conventions (e.g. CDC-48_UFD-1_NPL4 vs TIMELESS-TIPIN), which is confusing”***

Sorry for the confusion, which comes from the unique features of *C. elegans* terminology. Normally a hyphen would be used to link two associated proteins in a complex (e.g. TIMELESS-TIPIN), but something else is needed for *C. elegans* protein complexes, since individual protein names already contain a hyphen (e.g. CDC-48). Hence our use of an underscore to link components of a complex (thus CDC-48_UFD-1_NPL4). We have retained 'TIMELESS-TIPIN' at various points, since this is used in the literature as a generic name for these proteins in diverse eukaryotic species and so will be useful to the general reader who will not be familiar with *C. elegans* terminology.

- ***“The manuscript refers to Star methods and a Key Resources table. Neither of these appear to be part of the manuscript”***

Sorry about that and thanks for pointing this out. We now instead refer to 'Materials and Methods' and Appendix Table S1.

- ***“Line 400 refers to figure 3A-B. Shouldn't it be Figure 5A-B?”***

Yes – sorry about that and thanks for pointing this out. We have now corrected the mistake (line 447 of the revised manuscript).

- ***“Replisome' refers to all components except for CMG in Fig 2E, while in Fig 2B-D 'replisome' includes CMG”***

Thanks for pointing this out. For greater clarity, we have now listed the individual replisome factors in each figure panel.

- ***“Could the authors comment on the stoichiometry of the multi-subunit complexes they have purified (Figure 2A)?”***

The stoichiometry of the various multi-subunit complexes looks good. Remember that staining of proteins with Coomassie blue is proportional to size (bigger proteins stain more than smaller proteins). Therefore, the largest subunit of a stoichiometric complex should stain most heavily and the smallest will be stained the least. This is what we see in Figure 2A – the only complication is that some complexes (like CTF-18_RFC or the cullin ligases) have more than one subunit that migrate in a similar position in the gel, thus producing a band that stains more strongly.

Note that all multi-protein complexes were purified via a tag on a single subunit, before gel filtration. This means that we know that all the various subunits co-purify and co-migrate with each other.

- ***“Figure 2B is initially discussed (lines 205-224) without commenting on the enhanced ubiquitylation observed with LET-70 vs ARI-1 plus UBC-18. This difference is quite striking and should really be discussed as the data is introduced.”***

We have re-written this section (lines 215-237) to explain more clearly that LET-70 principally conjugates a single chain of up to ~8 ubiquitins to CMG-MCM-7 under these conditions (reactions that lack the E2 enzyme UBC-3). Such chains do not involve lysine 48 of ubiquitin (Figure EV1A, compare lanes 6 and 8). In contrast ARI-1 and UBC-18 mono-ubiquitylate CMG-MCM-7 on 1-3 sites under these conditions (Figure EV1A, compare lanes 2 and 3).

Therefore, both ARI-1_UBC-18 and LET-70 can prime ubiquitylation on CMG-MCM-7 (unlike UBC-3 or UBC-7), but UBC-3 or UBC-7 are then required to extend the K48-linked chains on CMG-MCM-7 that subsequently support helicase disassembly by CDC-48.

- ***“Does the K48-linked ubiquitylation on Lines 229-231 refer only to ubiquitylation initiated by UBC-18 and ARI1? The LET-70 ubiquitylation appears to be fairly independent of K48 (lanes 6 vs 9, Fig 2D).”***

With respect, the reviewer has confused ‘K48-only’ ubiquitin (ubiquitin in which K48 is intact but every other lysine has been mutated to arginine) with ‘K48R’ ubiquitin (ubiquitin in which K48 has been mutated to arginine but every other lysine is intact). The data in Figure 2D actually show that ubiquitylation of CMG-MCM-7, in the presence of the E2 enzymes LET-70 and UBC-3, is principally (though not exclusively) in the form of a single K48-linked ubiquitin chain (wt Ubi in lane 6 shows polyubiquitylation; K0 in lane 7 shows ubiquitylation is principally at one site per MCM-7 molecule; K48R in lane 8 shows that most ubiquitin chains require K48 of ubiquitin; K48-only in lane 9 shows that K48 is sufficient for chain formation under these conditions).

As discussed above, LET-70 primes ubiquitylation and on its own can form ubiquitin chains that do not involve K48 of ubiquitin. However, in the presence of

both LET-70 and UBC-3, LET-70 mono-ubiquitylates CMG-MCM-7 but UBC-3 then outcompetes LET-70 for chain formation, so that the dominant product is a single K48-linked ubiquitin chain.

- **“Lines 231-233 should more clearly explain how Figure 2B-C demonstrate that the major reaction product is a single polyubiquitin chain”**

We have modified the text as follows (lines 244-248):

“Moreover, the major product in reactions containing all three E2 enzymes, together with the E3 ligases CUL-2^{LRR-1} and ARI-1, is a single poly-ubiquitin chain on MCM-7 (since reactions containing lysine-free ubiquitin only support the conjugation of a single ubiquitin moiety to most MCM-7 molecules: compare Figure 2B lane 8 with Figure 2C lane 8).”

- **“Where is the UBC-1 and UBC-14 data referenced on lines 240-242?”**

The data are now shown in Figure EV1C and are cited on line 255-257.

- **“Neddylation of Cul2 as described on lines 243-246 should be clearly explained somewhere in the manuscript”**

The manuscript already explained that cullin neddylation is essential for the function of cullin ligases.

Lines 120-123 state that:

“the cullin scaffold must be modified by the ubiquitin-like protein NEDD8, which serves as a nexus that contacts multiple elements of the ligase along with the cognate E2 ubiquitin conjugating enzyme (Baek, Krist et al., 2020a, Wang et al., 2020).”

The subsequent sections then explain that the action of the priming E2 enzymes for cullin ligases is dependent upon cullin neddylation:

Lines 128-132:

“The first comprises paralogues of the E2 enzyme UBE2D, which is activated by the RING subunit of a neddylated cullin ligase (Baek et al., 2020a). The second priming enzyme is an RBR ligase of the ARIADNE family, known as ARIH1, which associates with neddylated cullin ligases”.

Finally, the manuscript cites past literature showing that neddylation of human CUL2 is required for function (lines 261-262).

- **“Lines 284-288 should make the connection between TIMELESS-TIPIN/CTF4/CLASPIN and the yeast proteins in Baretic et al (2020) clearer.”**

We have modified the text as follows (lines 300-304):

“The major partners of CMG within the replisome have been best defined in budding yeast (Baretic, Jenkyn-Bedford et al., 2020) and comprise Tof1-Csm3 (TIMELESS-TIPIN in mammals; TIM-1_TIPN-1 in C. elegans), Ctf4 (CTF4 / AND-1 / WDHD1 in mammals, CTF-4 in C. elegans) and Mrc1 (CLASPIN in mammals; CLSP-1 in C. elegans).”

- **“In Figure 3F it appears that PSF-3 is ubiquitinated. A band present immediately above the non-specific band in conditions 2 and 3 but not condition 1. This should be commented on.”**

We thank the reviewer for pointing this out. Having investigated this further, we have shown that the band in question is not PSF-3, as it can also be seen in the

absence of CMG (see lane 4 of panel C of 'Figure for Reviewer 2 (NOT FOR PUBLICATION)'). Therefore, the band represents cross-reactivity of the PSF-3 antibody with another component of the reconstituted reaction (the band is dependent upon the presence of ubiquitin and CUL-2^{LRR-1}). To avoid confusion, and since we know that the band in question is not a modified form of PSF-3, we have now marked it as 'non-specific' in Figure 3F of the revised manuscript.

Referee #3:

“Xia et al. present significant progress on how metazoa disassemble replisomes during replication termination. Important players of the pathway are identified and characterised by loss-of-function approaches in C elegans and their mechanistic roles investigated using extensive biochemical re-constitution. The authors also present a first step in re-constituting parts of the metazoan replisome from purified components. In addition, specific ubiquitylation by Cul2-LRR1 of Mcm7 and CMG extraction by p97 are re-constituted. These in vitro reactions could be key to advance our understanding in future studies of the molecular machines involved. The work also provides advance in our understanding of how the p97 segregase works (the specific advance presented on ubiquitylation and segregation is harder for me to judge due to my area of expertise).

The present study is timely. It is consistent with earlier work of the Labib lab and other labs in yeast, Xenopus and C elegans showing how CMG is disassembled during termination and in the subsequent mitosis. The present study extends and confirms important concepts suggested before mainly in yeast, but also presents unexpected aspects on mechanism and in vivo function.

The study presents an enormous amount of high-quality work. The data presented seems flawless. The conclusions are clear and sufficiently supported by the experimental evidence presented.

I strongly support acceptance of the manuscript upon addressing a few points.”

We thank the reviewer for his / her interest in our work and explain below how we have addressed the points raised.

Issues to be addressed:

- “The biochemistry appears to suggest a linear relationship of the action of Timeless-Tipin and UBXN3: CMG extraction depends on Timeless-Tipin-mediated ubiquitylation, which is a prerequisite for UBXN3-dependent CMG extraction by p97. The functional experiments in Fig 6 suggest synergistic effects of simultaneously inactivating both pathways, which rather fits a model where the action of Timeless-Tipin and UBXN3 occurs in parallel pathways. Please present an explanation for this potential contradiction.”

As the reviewer points out, synergistic effects upon combination of two mutations, such as the ‘synthetic lethal phenotype’ in Figure 6C upon combination of tim-1 RNAi and ubnx-3 RNAi, can sometimes be produced when two factors act in ‘parallel pathways’. However, although the notion of ‘linear pathways’ and ‘parallel pathways’ works well in some areas of biology, for example when studying signalling, this way of thinking breaks down in other contexts, for example when studying the biology of complex molecular machines (such as the replisome).

To give a simple analogy, consider a person riding a bike. The action of the person’s legs is ‘upstream’ of the movement of the pedals-and-chain that propel the wheels. If the person has injured legs but the bike is in great condition, they might

still be able to ride (though not perfectly). Similarly, a fit person might still ride a rusty bike with a very stiff chain-and-pedals. Put those two defects together and a person with injured legs might easily find that she / he cannot successfully ride the rusty old bike. This does not mean that the person's legs and the pedals-and-chain are in 'parallel pathways', it just illustrates that two independent defects in a process involving a complex machine can easily be additive (in this case the 'machine' is 'person plus bike').

We think that something similar is happening in our system. It's true that ubiquitylation of CMG is 'upstream' of disassembly of CMG, but it's still a very simple and logical fact that partial defects in ubiquitylation can be additive with partial defects in the disassembly machinery. Partial defects in either the ubiquitylation machinery or the disassembly machinery need not on their own block disassembly, but might easily mean that disassembly now becomes dependent upon optimal functioning of the unaffected part of the machinery. In our experiments, RNAi depletion of TIM-1 causes a partial defect in CMG ubiquitylation (shorter K48-linked chains on CMG-MCM-7), which makes disassembly dependent upon optimal functioning of the disassembly machinery. Similarly, RNAi depletion of UBXN-3 causes a partial defect in the disassembly machinery, which makes disassembly dependent upon optimal ubiquitylation of CMG. Put those two defects together and it's like the injured person on the rusty old bike – the process just doesn't work.

- “Re-constituted partial *C. elegans* replisomes are used to re-constitute replisome-dependent Mcm7 ubiquitylation and disassembly. A characterisation of the partial replisomes is required to be able to judge the approach used. Please provide such characterisation, for example by showing association of the replisome components with CMG. To my view, some (perhaps very basic) evidence will be important even if the authors plan to publish this *C. elegans* replisome system separately. One reason why this is important is that these replisomes were not formed by origin-dependent replication initiation, but in an "artificial" way by origin-independent assembly, as it has been done successfully for yeast replisomes by the O'Donnell lab, for example.”

We now show in Figure EV2A that the various replisome components (and CUL-2^{LRR-1}) can associate with CMG, using our purified proteins. We discuss these data in lines 304-308 of the revised manuscript.

- “Fig 3: From line 290 the authors state 'In contrast, removal of CTF-4 or CLSP-1 had no impact on the efficiency of CMG ubiquitylation" A control is required to show that CTF-4 and CLSP-1 associate with the replisome.”

As noted above, we now show in Figure EV2A that purified CTF-4 and CLSP-1 can associate with CMG. We discuss these data in lines 304-308 of the revised manuscript.

- “From line 310 the authors write "These data likely reflect a direct interaction of Cul-2-LRR1 with TIM-1-TIPN-1 in the context of the replisome". Please test and state whether an interaction of Cul-2-LRR1 and TIM-1-TIPN-1 occurs outside the context of full replisomes, which would be more direct evidence for a direct interaction.”

In the original version of our manuscript, we showed that CUL-2^{LRR-1} co-purifies with CMG, dependent upon TIM-1-TIPN-1 and drives the ubiquitylation not

only of CMG-MCM-7 (the preferred substrate in the replisome of CUL-2^{LRR-1}), but also of both subunits of TIM-1_TIPN-1 (Figure 3E-F plus what is now Figure EV2G). These data indicated that CUL-2^{LRR-1} has at least two important interactions with the replisome, one with TIM-1_TIPN-1 and another with CMG.

Many areas of biology are driven by multiple weak interactions that together provide sufficient affinity for the required biology, but that are not detectable in isolation. In the revised manuscript, we present glycerol gradient centrifugation data for mixtures of purified CMG, TIM-1_TIPN-1 and CUL-2^{LRR-1} (Figure EV2F). In this way, we observed that a fraction of CUL-2^{LRR-1} co-migrates with both CMG and TIM-1_TIPN-1, but only when all three proteins are mixed together. These data provide further evidence to support the idea that CUL-2^{LRR-1} interacts with both CMG and TIM-1_TIPN-1 in the context of the replisome.

Consistent with the view that CUL-2^{LRR-1} does indeed interact directly with TIM-1_TIPN-1, we also provide further new data to show that CUL-2^{LRR-1} can support the ubiquitylation of TIM-1_TIPN-1 in the absence of CMG and other replisome factors (Figure EV2H lane 3). However, ubiquitylation of TIM-1_TIPN-1 is more efficient in the presence of CMG, consistent with the fact that three factors form a ternary complex in the context of the replisome (Figure EV2H lane 1; lanes 2-3 show that this effect is specific to CMG, since other replisome factors do not impact on the efficiency of TIM-1_TIPN-1 ubiquitylation).

- ***“Line 463: "...likely reflecting the associated defect in CMG helicase disassembly during DNA replication termination”.***

I am not sure there is sufficient evidence for this conclusion. Can an alternative model be excluded where synthetic lethality results from defective replication termination-dependent disassembly in conjunction with a simultaneous defect in disassembly in the subsequent mitosis? Please elaborate or present this alternative explanation for the synthetic lethality.”

We now present new data in Figure EV5C to show that RNAi depletion of TIM-1 in *trul-1Δ* worms that lack mitotic CMG disassembly does not cause the dramatic loss of viability that is seen when *tim-1* RNAi is combined with *ubxn-3* RNAi (Figure 6C). Similarly, RNAi to *ubc-3+ubc-7* is synthetic lethal with *ubxn-3* RNAi (Figure 1F), and the same is true for combination of 10% *Irr-1* RNAi with *ubxn-3* RNAi (Figure 1D), yet similar lethality is not seen when *ubc-3+ubc-7* RNAi or 10% *Irr-1* RNAi is expressed in *trul-1Δ* worms (Figure EV5C). These data indicate that partial defects in the S-phase CUL-2^{LRR-1} pathway for CMG disassembly are synthetic lethal with *ubxn-3*, but are not synthetic lethal with loss of the mitotic TRUL-1-dependent pathway. Since full loss of LRR-1 is lethal, the simplest view of these data would be that the observed synthetic lethality in Figure 6C, Figure 1D and Figure 1F result from additive defects in the S-phase pathway for CMG disassembly during DNA replication termination (these defects ~phenocopy the exposure of worms to 100% *Irr-1* RNAi).

Minor issues

- ***“Introduction line 74 to line 95. This paragraph is hard to understand and should be revised.”***

We thank the reviewer for pointing this out and we have now revised the paragraph (lines 77-98).

- ***“line 112: typo: identify should be identity”***

We thank the reviewer for pointing this out – we have now corrected this typo (on what is now line 115).

- **“line 139: Is *C. elegans* *ubc-3* the only orthologue of human UBE2R1 and 2 (line 133)? Please make a statement.”**

C. elegans UBC-3 is indeed the only orthologue of human UBE2R1 and UBE2R2. We now make this clear on line 142-143.

- **line 168: “...and with other E2 enzymes.” How do you conclude this? Because the effect of *Ubc3* and *Ubc7* RNAi is not as strong as with *LRR1* siRNA? Please specify.”**

We now discuss on lines 173-177 of the revised manuscript that the combination of *ubc-3*Δ and *ubc-7*Δ is viable in *C. elegans* (data shown in Appendix Figure S1G), although brood size is reduced (data shown in Appendix Figure S1H). Therefore, other E2 enzymes must be able to act with CUL-2^{LRR-1} (since both *cul-2* and *lrr-1* are essential genes). LET-70 and UBC-18 are examples of such additional E2s.

- **“line 270/71: Specify the experiment made by changing to “However, the association of...that lack all other replisome components except CMG”**

We thank the reviewer for this suggestion and have now made the suggested change on lines 286-287 of the revised version of the manuscript.

- **“line 360: typo: *UBNX-3*”**

We thank the reviewer and have now made the correction on lines 402 of the revised version of the manuscript.

A
'Mix of proteins expressed in yeast'

UBA-1
 POL ϵ
 CLSP-1
 CTF-18_RFC
 TIM-1_TIPN-1
 CTF-4
 ARI-1
 ULA-1_RFL-1

'Mix of proteins expressed in bacteria'

MCM-10
 LET-70
 UBC-18
 UBC-3
 UBC-12
 DCN-1
 NED-8
 Ub-K0
 Ub-K48only
 Ub-K48R

B

-	-	-	-	+	+	+	+	Ub-Prg (deubiquitylase inhibitor)
-	-	-	+	-	-	-	+	Human USP2 deubiquitylase
-	+	-	-	-	+	-	-	Mix of proteins expressed in yeast
-	-	+	-	-	-	+	-	Mix of proteins expressed in bacteria

Immunoblot
 to detect
 poly-ubiquitin chains
 attached to MCM-7
 (FK2 antibody)

1 2 3 4 5 6 7 8
 * = non-specific band

C

TIM-1_TIPN-1 + CLSP-1 +
 POL ϵ + CTF-18_RFC +
 CTF-4 + MCM-10 +
 E1 + E2s (LET-70; UBC-3; UBC-18)
 + ARI-1 + Neddylation

-	+	-	+	Ub
+	+	-	-	CMG
-	+	-	+	CUL-2 ^{LRR-1}

← the band that Reviewer 2 asked about

PSF-3-

1 2 3 4

Thank you for submitting your revised manuscript to The EMBO Journal. We have now heard back from the three original referees, and I am pleased to say that they all found the previously-raised points satisfactorily addressed. Following a final revision round to address some remaining minor issues noted by reviewers 2 and 3, as well as the below-listed editorial points, we shall therefore be happy to accept the study for publication in our journal.

Referee #1:

The authors have addressed all my concerns and significantly improved the manuscript. I encourage the publication of this manuscript.

Referee #2:

Xia and Colleagues have done a great job responding to reviewers' concerns. Figure EV5 is a particularly nice addition. I have no major concerns but my minor concerns (below) should be addressed with changes to the text before publication:

- The DUB control in panels A-B of "Figure for Reviewer 2" is an important one. This should be included somewhere in the supplement for publication.

- The authors misinterpreted my original request to clarify the neddylation described on lines 243-246 of the original manuscript. They really just need to clearly explain how the neddylation reaction was performed, which is currently not clear in Fig EV1D-F. In the revised manuscript it seems that the components used for neddylation are described on lines 858-859. These proteins used should be listed in the legend of Fig EV1D-F for clarity and ideally should also be defined as 'neddylation enzymes' in the main text.

Referee #3:

I pointed out the high relevance of the research presented by Xia et al and the high experimental quality in my review of the initially submitted manuscript.

The revisions presented by the authors in the revised manuscript support the conclusions even more and also improve readability of the very complex paper.

I fully support publication of the revised manuscript.

Minor point:

In Fig 5B, Ctf4 and Claspin are decreased upon tim-1 RNAi treatment. Earlier in the manuscript, the stability of Ctf4 and Claspin association with recombinant CMG in the presence and absence of TIMELESS-TIPIN are not addressed. This raises the possibility that the lower Lrr1 association in the absence of TIMELESS is partly due to decreased Ctf4 and Claspin, although they are shown to be individually dispensable for LRR1 binding/Mcm7 ubiquitylation. This issue does not change that main conclusion, of course, that TIMELESS-TIPIN is the main factor. But because Ctf4 and Mrc1 are important in yeast the effect on these factors in *C. elegans* is worth mentioning.

Once again, we thank all three reviewers for their comments and for their interest in our work. The remaining issues that were raised by the reviewers are discussed below.

Referee #1:

“The authors have addressed all my concerns and significantly improved the manuscript. I encourage the publication of this manuscript.”

Referee #2:

“Xia and Colleagues have done a great job responding to reviewers' concerns. Figure EV5 is a particularly nice addition. I have no major concerns but my minor concerns (below) should be addressed with changes to the text before publication:”

- ***“The DUB control in panels A-B of "Figure for Reviewer 2" is an important one. This should be included somewhere in the supplement for publication.”***

As suggested by the editor, the “Figure for Reviewer 2” will be retained in the Review Process File as part of the point-by-point response.

- ***“The authors misinterpreted my original request to clarify the neddylation described on lines 243-246 of the original manuscript. They really just need to clearly explain how the neddylation reaction was performed, which is currently not clear in Fig EV1D-F. In the revised manuscript it seems that the components used for neddylation are described on lines 858-859. These proteins used should be listed in the legend of Fig EV1D-F for clarity and ideally should also be defined as 'neddylation enzymes' in the main text.”***

The 'neddylation enzymes' are illustrated and labelled in Figure 2A. We now list these enzymes (ULA-1_RFL-1, UBC-12 and DCN-1) in the relevant section of the main text (lines 220-221). In all the relevant figure legends, we now also explain that “Neddylation” in the corresponding figure panels indicates addition to the reactions of the *C. elegans* ULA-1_RFL-1 E1 enzyme, the UBC-12 E2 enzyme, the DCN-1 E3 enzyme and NED-8.

Referee #3:

“I pointed out the high relevance of the research presented by Xia et al and the high experimental quality in my review of the initially submitted manuscript. The revisions presented by the authors in the revised manuscript support the conclusions even more and also improve readability of the very complex paper. I fully support publication of the revised manuscript.

Minor point: In Fig 5B, Ctf4 and Claspin are decreased upon tim-1 RNAi treatment. Earlier in the manuscript, the stability of Ctf4 and Claspin association with recombinant CMG in the presence and absence of TIMELESS-TIPIN are not addressed. This raises the possibility that the lower Lrr1 association in the absence of TIMELESS is partly due to decreased Ctf4 and Claspin, although they are shown to be individually dispensable for LRR1 binding/Mcm7 ubiquitylation. This issue does not change that main conclusion, of course, that TIMELESS-TIPIN is the main factor. But because Ctf4 and Mrc1 are important in yeast the effect on these factors in C elegans is worth mentioning.”

It's true, as the referee mentions, that the association of CTF-4 and CLSP-1 with the worm replisome is somewhat destabilised in the absence of TIM-1 (Figure 5).

However, CMG ubiquitylation and the association of CUL-2^{LRR-1} with CMG are not affected by depletion of CLSP-1 in *ctf-4Δ* worms (Figure EV4C, discussed on lines 454-460). Moreover, we show in Figure EV2F that CMG, CUL-2^{LRR-1} and TIM-1_TIPN-1 form a ternary complex in the absence of CTF-4 and CLSP-1. The association of CUL-2^{LRR-1} with CMG in this reconstituted complex is dependent upon the presence of TIM-1_TIPN-1 (in the complete absence of CTF-4 and CLSP-1).

Therefore, it's clear that CTF-4 and CLSP-1 cannot explain the role of TIM-1_TIPN-1 in recruiting CUL-2^{LRR-1} to CMG and promoting efficient CMG ubiquitylation.

Thank you for submitting your final revised manuscript for our consideration. I am pleased to inform you that we have now accepted it for publication in The EMBO Journal.

Corresponding Author Name: KARIM LABIB

Journal Submitted to: The EMBO Journal

Manuscript Number: EMBOJ-2021-108053